# Molecular mechanism for the control of virulent *Toxoplasma gondii* infections in wild-derived mice

Mateo Murillo-León [1,2,3], Urs B. Müller[4], Ines Zimmermann [1,3], Shishir Singh [1,2,3], Pia Widdershooven[4,5], Cláudia Campos [6], Catalina Alvarez [6], Stephanie Könen-Waisman[7], Nahleen Lukes [8], Zsolt Ruzsics[1,2], Jonathan C. Howard [6], Martin Schwemmle[1,2] & Tobias Steinfeldt[1,2]

Some strains of the protozoan parasite *Toxoplasma gondii* (such as RH) are virulent in laboratory mice because they are not restricted by the Immunity-Related GTPase (IRG) resistance system in these mouse strains. In some wild-derived Eurasian mice (such as CIM) on the other hand, polymorphic IRG proteins inhibit the replication of such virulent *T. gondii* strains. Here we show that this resistance is due to direct binding of the IRG protein Irgb2-b1$_{CIM}$ to the *T. gondii* virulence effector ROP5 isoform B. The Irgb2-b1 interface of this interaction is highly polymorphic and under positive selection. South American *T. gondii* strains are virulent even in wild-derived Eurasian mice. We were able to demonstrate that this difference in virulence is due to polymorphic ROP5 isoforms that are not targeted by Irgb2-b1$_{CIM}$, indicating co-adaptation of host cell resistance GTPases and *T. gondii* virulence effectors.

[1] Institute of Virology, Medical Center University of Freiburg, 79104 Freiburg, Germany. [2] Faculty of Medicine, University of Freiburg, 79104 Freiburg, Germany. [3] Faculty of Biology, University of Freiburg, 79104 Freiburg, Germany. [4] Institute for Genetics, University of Cologne, 50674 Cologne, Germany. [5] Department of Biology, University of Cologne, 50674 Cologne, Germany. [6] Fundação Calouste Gulbenkian, Instituto Gulbenkian de Ciencia, 2780-156 Oeiras, Portugal. [7] Department for Dermatology and Venereology, University Hospital of Cologne, 50937 Cologne, Germany. [8] Institute of Immunology, University Hospital Aachen, 52074 Aachen, Germany. These authors contributed equally: Mateo Murillo León, Urs B. Müller. Correspondence and requests for materials should be addressed to T.S. (email: tobias.steinfeldt@uniklinik-freiburg.de)

In the co-evolutionary process of host–pathogen interaction, adaptation to local conditions is critical. Infectious agents are a constant threat to multicellular organisms, and all metazoan organisms have evolved immune defense mechanisms to combat virulent microbes. Immune defense mechanisms emerge from selective pressures that microbes impose; invasive microbes, in turn, evolve to avoid or counteract immune effector mechanisms long enough to allow for efficient transmission. The host and the pathogen undergo co-adaptation at the molecular level. These equilibria are unstable and their conditions vary locally.

*Toxoplasma gondii* is an obligate intracellular protozoan parasite belonging to the phylum Apicomplexa. It is distantly related to the genus *Plasmodium*, the causative agent of malaria. Unlike *Plasmodium*, however, *T. gondii* has an extraordinarily broad host range, with all true cats (Felidae) as definitive hosts and all warm-blooded animals, including birds and humans, as intermediate hosts. About one-fourth of the human population is infected with *T. gondii*, although local rates vary considerably[1]. A few lineages predominate in Europe and North America[2–4] and these canonical strains differ markedly in virulence in laboratory strains of mice. Virulent strains (e.g., restriction fragment length polymorphism (RFLP) genotype 10, previously called type I) are usually lethal following inoculation with even a single parasite, whereas the $LD_{50}$ (lethal dose, 50%) of avirulent strains (e.g., RFLP genotypes 1, 2, 3, previously called types II and III) ranges between $10^3$ and $10^5$ in laboratory mice[5,6]. These virulence differences are due to a small number of polymorphic genes within the parasite genomes[7–9]. However, in other parts of the world, genetically highly diverse *T. gondii* strains have been isolated[4,10–14], and especially in South America the majority of isolates is associated with high mortality rates in laboratory mice[11,15].

Immunity-Related GTPases (IRG) drive an essential mechanism of early cell-autonomous resistance against intracellular pathogens in mice[16]. IRG proteins are represented by about 20 single coding units in the C57BL/6 (BL/6) mouse genome[17] and multiple members are highly inducible by interferon-γ (IFNγ)[18–20]. The typical *IRG* gene has one or two short 5'-untranslated exons followed by a single long exon encoding the entire open reading fame. Four *IRG* genes depart from this structure, with two adjacent long exons each encoding a full-length IRG polypeptide, resulting in expression of proteins carrying two IRG domains joined by a short linker, subtending a single promoter, the so-called tandem IRG proteins[17,21].

Knockout (ko) mouse strains lacking single or multiple IRG members have consistently shown higher susceptibility to infection with normally avirulent *T. gondii* strains[22–24]. Following infection of an IFNγ-stimulated cell of a laboratory mouse, guanosine triphosphate (GTP)-activated effector IRG proteins begin to accumulate at the *T. gondii* parasitophorous vacuolar membrane (PVM) within minutes after invasion[25–27]. Premature activation in uninfected cells is prevented by the three regulator IRG proteins, Irgm1, 2 and 3, which keep the effector IRG proteins in a guanosine diphosphate (GDP)-bound inactive conformation at endogenous cellular membranes[28,29] until infection. The loading of effector IRG proteins is cooperative and hierarchical, with two family members serving as pioneers for members loading later in the hierarchy[26]. IRG protein accumulation is a prerequisite for subsequent disruption of the PVM[27,30–32], which is in turn invariably followed by death of the parasite and subsequent necrotic death of the host cell[33]. A mechanochemical effector function of IRG proteins is suggested, by analogy to the dynamins[19,34], to be responsible for the ruffling, vesiculation and ultimately disruption of the PVM observed at the microscopic level[27,30,32].

Virulence of *T. gondii* can be directly correlated with inactivation of the IRG resistance system. The initial loading of the PVM with IRG proteins is markedly reduced in virulent compared with avirulent strains[26]. To preserve the integrity of the PVM, *T. gondii* has evolved several polymorphic virulence effectors that are injected from secretory organelles (rhoptries and dense granules) directly into the cytosol during and after host cell invasion[1,35]. Genetic screens showed that the polymorphism in the rhoptry-derived ROP5 and ROP18 components of a secreted threonine kinase complex accounts for the differences in virulence between *T. gondii* strains in infected laboratory mice[5,36,37]. Several *T. gondii* effectors have been shown to inactivate mouse effector IRG proteins, thus preserving integrity of the PVM[38–45].

Recently, we showed considerable polymorphism in IRG proteins among several wild and wild-derived inbred mouse strains. The wild-derived *Mus musculus castaneus* strain CIM from South India counters effectors of Eurasian virulent strains, leading to encystment, and therefore potential transmission, of virulent parasites. In breeding experiments the resistance of CIM mice could be mapped to highly polymorphic *IRG* genes located on chromosome 11. Within this locus, one of the most polymorphic family members is the tandem IRG protein Irgb2-b1. We could show that in transiently transfected BL/6 cells, overexpression of Irgb2-b1$_{CIM}$ can rescue the effector IRG protein, Irga6$_{BL/6}$, from ROP5/ROP18/GRA7-mediated phosphorylation[21]. However, these observations did not show that protection of Irga6 is sufficient to enable wild-derived CIM mice themselves to fully resist infection by *T. gondii* virulent strains. Indeed, in BL/6 mice, resistance is only partially lost after deletion of *Irga6*$_{BL/6}$[23], suggesting that in the case of CIM mice, protection of Irga6$_{CIM}$ by Irgb2-b1$_{CIM}$ may not be sufficient to explain full resistance against virulent Eurasian *T. gondii* strains.

In the present study, we show that Irgb2-b1$_{CIM}$ is the CIM-inherited element largely responsible for resistance against virulent Eurasian *T. gondii* strains. We demonstrate efficient binding of Irgb2-b1$_{CIM}$ only to virulent *T. gondii*-derived ROP5B but not ROP5A and ROP5C. The interface necessary for ROP5B binding is located within the N-terminal portion of Irgb2-b1$_{CIM}$ and encompasses structural motifs that were previously shown to be under positive selection. Our findings are consistent with critical responsibility of ROP5 for the heightened virulence of *T. gondii* type I strains against laboratory *Mus musculus*, and inevitably suggest that other ROP pseudokinases may not play a significant role in virulence/avirulence behavior in mice against *T. gondii*. Furthermore, and supporting this last contention, we provide evidence that virulence of *T. gondii* strains from South America in CIM mice is due to a mismatch between Irgb2-b1$_{CIM}$ and ROP5 isoforms. These results provide further evidence that *T. gondii* virulence and mouse resistance follow some form of allele-matching evolutionary dynamics.

## Results

**Irgb2-b1$_{CIM}$ protects from virulent *T. gondii* strain infections**. Our own recent work demonstrates that the highly polymorphic *Irgb2-b1*$_{CIM}$ tandem gene may be responsible for resistance of wild-derived CIM mice against virulent Eurasian *T. gondii* strains[21]. In transiently transfected IFNγ-induced BL/6 cells, expression of Irgb2-b1$_{CIM}$ protected endogenous Irga6$_{BL/6}$ from ROP5/ROP18/GRA7-mediated phosphorylation by the virulent strain RH. High levels of Irgb2-b1$_{CIM}$ at the PVM of *T. gondii* virulent strains were accompanied by absence of Irga6$_{BL/6}$ phosphorylation[21]. To test definitively the relevance of Irgb2-b1$_{CIM}$ for the resistance of CIM cells against virulent *T. gondii*, we generated two independent lines of *Irgb2-b1*$_{CIM}$ ko cells (T17 and i3) by CRISPR/Cas9 (clustered regularly interspaced short palindromic repeats/CRISPR-associated protein 9) nickase technology[46]. Loss of *Irgb2-b1*$_{CIM}$ was confirmed in lysates of

IFNγ-stimulated cells by western blot with an antibody specific for the C terminus of tandem IRG proteins (Fig. 1a)[21] and Irgb2-b1$_{CIM}$ could not be detected by immunofluorescence on vacuoles of RHΔ*hxgprt* strain *T. gondii* in ko cells (Fig. 1b, representative images are shown in Supplementary Fig. 1). Expression levels of IRG effector proteins Irga6, Irgb6, Irgb10 and Irgd, necessary for full resistance against avirulent strains of *T. gondii* in laboratory mice[16], were not found to be different in CIM wild-type (wt) cells compared with Irgb2-b1$_{CIM}$ ko cells (Fig. 1a).

Phosphorylation of Irga6$_{BL/6}$ in susceptible BL/6 cells by the ROP5/ROP18/GRA7 kinase complex of virulent *T. gondii* strains occurs mainly at two threonine residues (T102 and T108) in the G-domain[39,40]. In wt CIM cells, Irga6$_{CIM}$ phosphorylation by RHΔ*hxgprt* strain *T. gondii* is almost completely inhibited (Fig. 1c); however, in the two Irgb2-b1$_{CIM}$ ko CIM cell lines, the frequencies of vacuoles carrying pT108-phosphorylated Irga6$_{CIM}$ are comparable to frequencies in BL/6 cells (Fig. 1c, representative images are shown in Supplementary Fig. 2). The ability of IFNγ-induced CIM wt and Irgb2-b1$_{CIM}$ ko cells to control replication of RH-YFP was compared in infected cells by flow cytometry (Fig. 1d, gating strategy and representative images are shown in Supplementary Fig. 3) and incorporation of ³H-uracil (Supplementary Fig. 4). In both assays, Irgb2-b1$_{CIM}$ ko cells lost the ability to restrict *T. gondii* virulent strain replication shown by CIM wt cells. The Irgb2-b1$_{CIM}$ ko cell line T17 was complemented with Irgb2-b1$_{CIM}$ and Irgb2-b1$_{CIM}$ expression was confirmed by western blot of detergent lysates (Supplementary Fig. 5). In unstimulated cells, Irgb2-b1$_{CIM}$ expression is only detectable in complemented but not wt cells (left hand panel), whereas upon stimulation with IFNγ, Irgb2-b1$_{CIM}$ levels are similar in CIM wt and complemented cells (right hand panel). Low Irgb2-b1 expression levels in BL/6 wt cells have been described earlier[21], and Irgb2-b1$_{CIM}$ ko cells show no signs of Irgb2-b1$_{CIM}$ expression. Complementation of ko cells with Irgb2-b1$_{CIM}$ restored resistance against virulent *T. gondii* RH-YFP to wt levels (Fig. 1e).

**Irgb2-b1$_{CIM}$ is mainly associated with ROP5B.** Several published results have demonstrated a specific molecular interaction between the ROP5 pseudokinase component of the ROP5/ROP18/GRA7 kinase complex of virulent Eurasian *T. gondii* strains, and the IRG effector, Irga6$_{BL/6}$[38,41,43,47]. A structurally similar interaction has been proposed between Irgb2-b1$_{CIM}$ and ROP5 based on the predicted architecture shared by all IRG proteins[48]. In contrast to other IRG proteins, ROP5 of the mouse virulent strain RH is required for vacuolar loading of Irgb2-b1$_{CIM}$ and variation in ROP5 expression levels of different *T. gondii* RH sub-strains correlate directly with the amount of vacuolar Irgb2-b1$_{CIM}$ loading. In CIM cells, vacuoles of *rop5*-deficient RH load very little Irgb2-b1$_{CIM}$[21]. A plausible explanation is that virulent ROP5 bound to the vacuolar membrane is a direct target for Irgb2-b1$_{CIM}$.

We here demonstrate an Irgb2-b1$_{CIM}$/ROP5 interaction by co-immunoprecipitation from detergent lysates of CIM cells infected with the *T. gondii* virulent RHΔ*hxgprt* strain (Fig. 2a, upper left hand panel). Protein amounts in the lysates used for co-immunoprecipitation are shown in the middle and right hand panels. This interaction could be confirmed by pull-down of ROP5 from *T. gondii* tachyzoite detergent lysates with a purified Irgb2-b1$_{CIM}$ glutathione *S*-transferase (GST)-fusion protein. When the BL/6 variant of Irgb2-b1 was used, no binding to ROP5 was detectable under similar conditions (Fig. 2b, middle left hand panel). Equal protein amounts and validation of *T. gondii* strains used in the pull-down were confirmed with tachyzoite lysates (Fig. 2b, right hand panels). In both

experiments (Fig. 2a, b) appearance of GRA7 in the pull-down is consistent with a direct interaction with ROP5, as previously demonstrated[43].

These data demonstrate that recruitment of Irgb2-b1$_{CIM}$ to the PVM of *T. gondii* virulent RHΔ*hxgprt* strain is dependent on ROP5. However, they neither prove a direct binary interaction nor discriminate between distinctive ROP5 isoforms A, B and C.

The yeast two-hybrid (Y2H) assay allows direct screening for interaction of two proteins. Therefore, Irgb2-b1$_{CIM}$ and virulent RHΔ*hxgprt T. gondii* isoforms ROP5A/B/C were expressed as N-terminal fusion proteins with the Gal4 DNA-binding (BD) or Gal4 activation domain (AD) in a yeast reporter strain. Colony growth on selective medium indicated direct interaction of Irgb2-b1$_{CIM}$ with ROP5B. Significantly less colony growth was observed for ROP5C, and no interaction with ROP5A was detectable (Fig. 2c). In the protein-fragment complementation assay (PCA), binding of Irgb2-b1$_{CIM}$ to ROP5 isoforms B and C could be reproduced and, also in this case, binding to ROP5B was significantly stronger than to ROP5C (Fig. 2d).

**The Irgb2-b1$_{CIM}$ interface for ROP5B binding is under positive selection.** Many IRG proteins are highly polymorphic in *Mus musculus*, but only the N-terminal unit of the Irgb2-b1 tandem, Irgb2, has been under recent positive selection[21]. A significant evolutionary hotspot encompassing putative structural motifs αd and H4 spans Irgb2 nucleotides 500 to 700. Both αd and H4 constitute essential parts of the Irga6$_{BL/6}$ interface for ROP5 binding[38,47] and we considered that these motifs probably also participate in Irgb2-b1$_{CIM}$ binding to ROP5. Initially, the N-terminal moiety Irgb2$_{CIM}$ and C-terminal moiety Irgb1$_{CIM}$ were expressed separately as GST-fusion proteins and applied in a pull-down approach with *T. gondii* virulent tachyzoite detergent lysates. Binding of Irgb2$_{CIM}$ but not Irgb1$_{CIM}$ to *T. gondii*-derived ROP5 (Fig. 3a) strongly suggests that the interface for ROP5 binding is located in the polymorphic N-terminal part of Irgb2-b1$_{CIM}$. Input of GST-fusion proteins (Fig. 3a, upper left hand panel) and protein amounts in the tachyzoite lysate used in the pull-down (Fig. 3a, right hand panels) is shown. To demonstrate that αd and H4 of Irgb2$_{CIM}$ are responsible for ROP5 recognition, a chimeric Irgb2-b1 protein was generated and tested for ROP5 binding. The Irgb2-b1$_{Chimera}$ encompasses a modified Irgb2$_{CIM}$, where αd and H4 were replaced with the respective BL/6 sequences, followed by full-length Irgb1$_{BL/6}$ (Fig. 3b).

In the PCA, binding of the Irgb2-b1$_{Chimera}$ to ROP5B was almost completely abrogated, whereas binding to ROP5C was even more pronounced than wt Irgb2-b1$_{CIM}$ binding to ROP5C (Fig. 3c). These results were confirmed in a Y2H approach. Only ROP5C but not ROP5B binding to Irgb2-b1$_{Chimera}$ could be observed (Fig. 3d). Binding of Irgb2-b1$_{Chimera}$ to ROP5C was also reflected in recruitment to the *T. gondii* virulent strain-derived PVM. After complementation of Irgb2-b1$_{CIM}$ ko cells, the vacuolar intensities of Irgb2-b1$_{Chimera}$ are significantly lower than Irgb2-b1$_{CIM}$, but still detectable (Fig. 3e). We could not observe differences in numbers (protein intensities are not considered in these analyses) of Irgb2-b1$_{CIM}$- and Irgb2-b1$_{Chimera}$-positive vacuoles (Fig. 3f) confirming the association of both proteins with different ROP5 isoforms.

**ROP5B is the main isoform responsible for *T. gondii* virulence in vitro.** Binding of Irgb2-b1$_{Chimera}$ to ROP5C stimulated us to reinvestigate binding of Irgb2-b1$_{BL/6}$ to virulent strain-derived ROP5 isoforms. In the PCA, only binding of Irgb2-b1$_{BL/6}$ to ROP5C, but not to ROP5B or ROP5A, could be detected (Fig. 4a); these results were further confirmed by Y2H analysis (Fig. 4b). Surprisingly, Irgb2-b1$_{BL/6}$ binding to ROP5C could not be

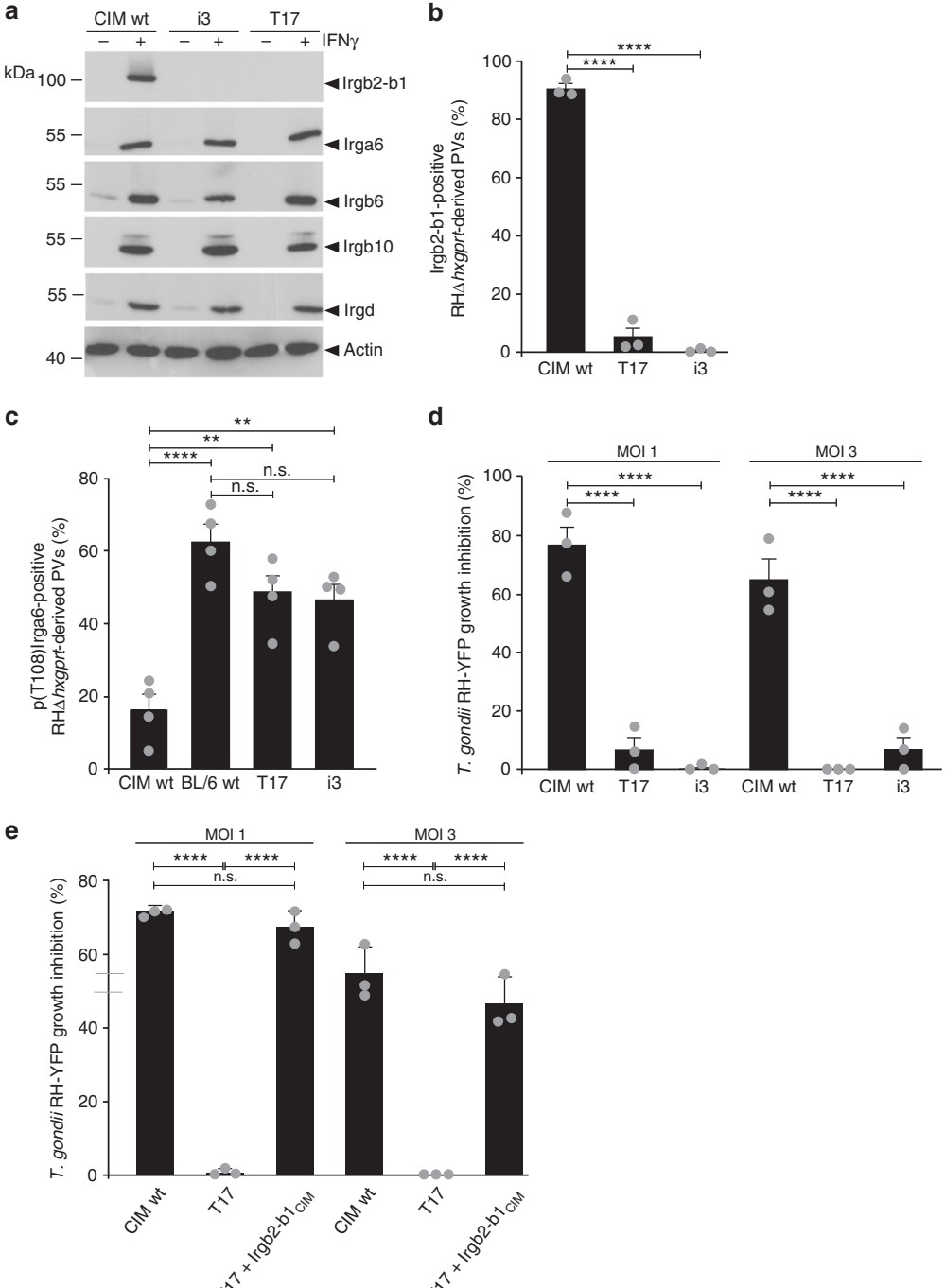

**Fig. 1** Irgb2-b1$_{CIM}$ is largely responsible for resistance against virulent *T. gondii* strains. **a** Western blot of detergent lysates from CIM wild-type (wt) and *Irgb2-b1*$_{CIM}$ knockout (ko) diaphragm-derived cells (DDCs) (T17 and i3) stimulated for 24 h with 200 U ml$^{-1}$ interferon-γ (IFNγ). The signal representing Irgb2-b1$_{CIM}$ in wt cells is lost in both ko cell lines (upper panel). Expression levels of all other Immunity-Related GTPases (IRG) proteins are unchanged in ko cell lines compared with wt cells (middle panels). Actin serves as loading control (lower panel). **b, c** Frequency of vacuoles positive for Irgb2-b1$_{CIM}$ (**b**) or Irga6$_{CIM}$ phosphorylated at T108 p(T108)Irga6$_{CIM}$ (**c**) detected by immunofluorescence with anti-p(T108)Irga6$_{CIM}$ or anti-Irgb2-b1$_{CIM}$ antibodies in IFNγ-induced (200 U ml$^{-1}$) wt CIM and BL/6 or *Irgb2-b1*$_{CIM}$ ko CIM DDCs infected for 2 h with RHΔ*hxgprt*. Error bars indicate the mean and standard error of the mean (SEM) of three (**b**) or four (**c**) independent experiments (about 100 vacuoles were counted per experiment). One-way analysis of variance (ANOVA) followed by Tukey's multiple comparison was used to test differences between groups; ****$p < 0.0001$; **$p < 0.0025$; n.s. not significant. **b** No Irgb2-b1$_{CIM}$ positive vacuoles were found in *Irgb2-b1*$_{CIM}$ ko cells. **c** Frequencies of p(T108)Irga6$_{CIM}$-positive vacuoles in ko cells are significantly increased in comparison with wt cells. **d, e** CIM DDCs were induced with 100 U ml$^{-1}$ IFNγ for 24 h and infected with *T. gondii* RH-YFP at a multiplicity of infection (MOI) of 1 or 3. Intracellular parasite growth was determined by flow cytometry 24 h post infection as described in Methods. Error bars indicate the mean and SEM of three independent experiments. One-way ANOVA followed by Tukey's multiple comparison was used to test differences between groups; ****$p < 0.0001$; n.s. not significant. **d** Control of *T. gondii* replication is lost in wt BL/6 and *Irgb2-b1*$_{CIM}$ ko CIM DDCs compared with wt CIM DDCs. **e** Control of *T. gondii* replication is restored to wt levels (wt CIM DDCs) in *Irgb2-b1*$_{CIM}$ ko CIM DDCs complemented with *Irgb2-b1*$_{CIM}$ (T17+Irgb2-b1$_{CIM}$)

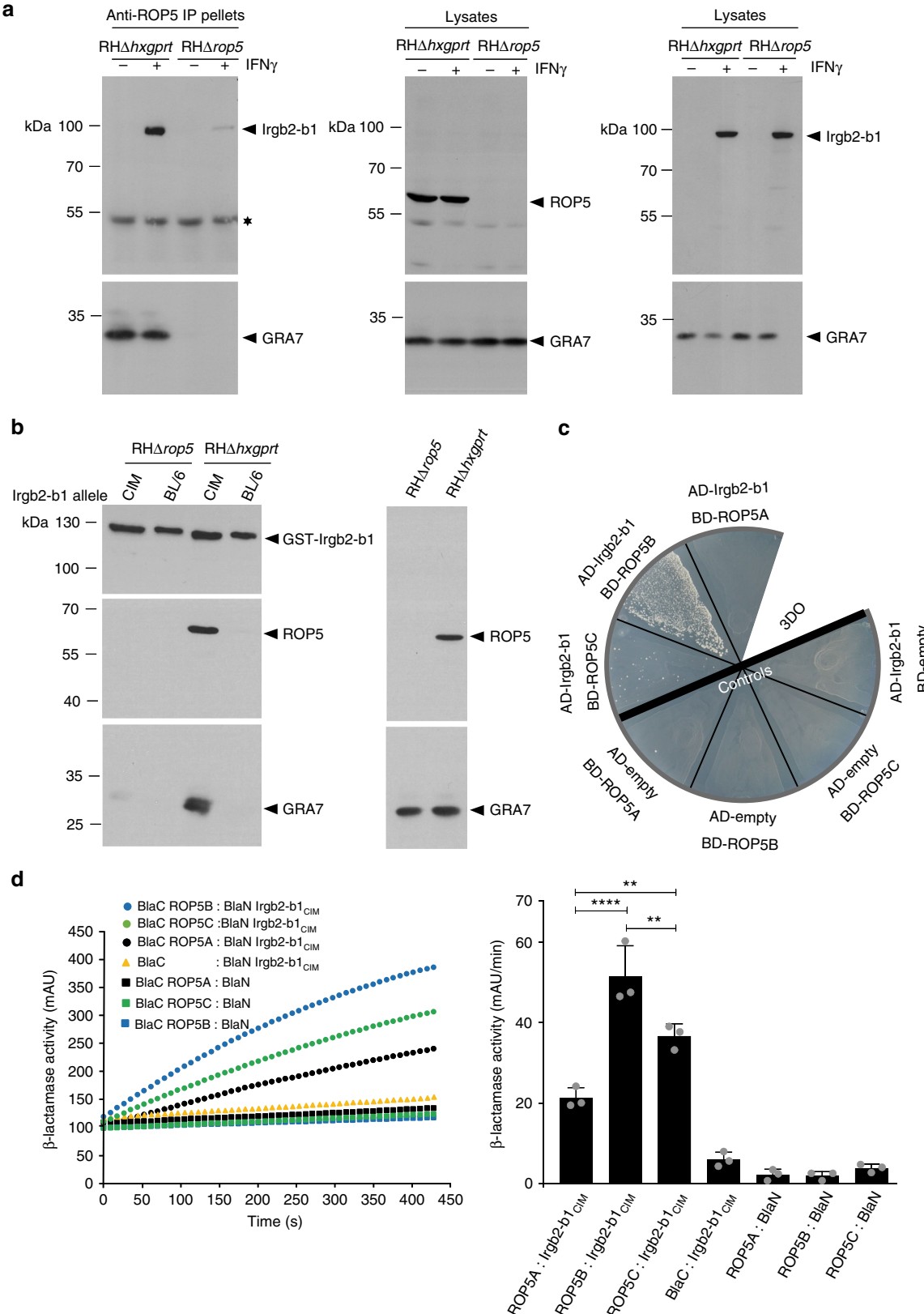

detected in the pull-down with a GST-tagged fusion protein (Fig. 2b) but this proved to be due to discrimination between ROP5 isoforms by the antibody used in subsequent western blot analysis since, after overexpression of single FLAG-tagged ROP5 isoforms in transiently transfected cells, the anti-ROP5-specific

antibody 3E2 detects ROP5A and ROP5B but not ROP5C in the western blot of detergent cell lysates. Protein expression was verified with an anti-FLAG antibody, with calnexin levels serving as loading control (Supplementary Fig. 6). Phylogenetic analysis of *Irgb2* sequences[21] reveals the allelic diversity between

**Fig. 2** Irgb2-b1$_{CIM}$ directly binds *T. gondii*-derived ROP5B and ROP5C. **a** Irgb2-b1$_{CIM}$ was co-imunoprecipitated from RHΔ*hxgprt*- but not RHΔ*rop5*-infected CIM diaphragm-derived cells (DDCs) that have been induced with 200 U ml$^{-1}$ interferon-γ (IFNγ) using a ROP5-specific antibody (upper left hand panel). The lower left hand panel indicates GRA7 association with ROP5 immunoprecipitations. The star indicates a protein unspecifically detected by the antibody used for immunoprecipitation. The middle and right hand panels display amounts of ROP5, GRA7 and Irgb2-b1 in the lysates used for immunoprecipitation. **b** Pull-down of ROP5 by CIM but not BL/6 GST-Irgb2-b1 from RHΔ*hxgprt* detergent lysates. RHΔ*rop5* tachyzoite detergent lysates have been included as control (middle left hand panel). The lower panel indicates GRA7 association with ROP5. The upper panel indicates input of glutathione *S*-transferase (GST)-fusion proteins in the pull-down. The right hand blot shows ROP5 (upper panel) and GRA7 (lower panel) levels in tachyzoite lysates. **c** Irgb2-b1$_{CIM}$ interacts with ROP5B and ROP5C but not ROP5A in a yeast two-hybrid approach. Proteins were expressed either as fusion to a transcriptional activation domain (AD) from pGAD-C3 or to a DNA-binding domain (BD) from pGBD-C3. Colony growth under 3DO conditions is indicative of protein/protein interaction. The bold black line separates samples from negative controls. **d** Protein-fragment complementation assay. Proteins were fused to N-terminal (BlaN) or C-terminal (BlaC) fragments of the reporter protein *TEM-1* β-lactamase. The increase in absorbance measured at 405 nm indicates restoration of β-lactamase activity after protein/protein interaction. Binding of Irgb2-b1$_{CIM}$ to ROP5B is significantly stronger compared to ROP5C and ROP5A. Error bars indicate the mean and standard deviation of three independent experiments (right hand panel). One-way analysis of variance (ANOVA) followed by Tukey's multiple comparison was used to test differences between groups; ****$p < 0.0001$; **$p < 0.0025$. The kinetic of the β-lactamase reaction is shown for one representative experiment (left hand panel)

---

laboratory and wild-derived mice (Fig. 4e, upper panel). An alignment of amino acids 167 to 233 encompassing putative structural motifs αd and H4 highlight the polymorphic residues within this region (Fig. 4e, lower panel). To confirm the different binding specificities for ROP5B in infected cells, we used RHΔ*rop5* parasites expressing virulent allelic isoforms ROP5A and ROP5B with HA- or FLAG-epitope tag (RHΔ*rop5*+A/B expressing HA-ROP5A and FLAG-ROP5B)[49]. We can demonstrate ROP5B binding to Irgb2-b1$_{CIM}$ but not Irgb2-b1$_{BL/6}$ or Irgb2-b1$_{Chimera}$ by co-immunoprecipitation from detergent lysates of cells infected with *T. gondii* RHΔ*rop5*+A/B (Supplementary Fig. 7) again suggesting that residues within this polymorphic hotspot determine the interaction. Protein amounts in the lysates used for co-immunoprecipitation are shown (Supplementary Fig. 7, middle and right hand panels).

Irgb2-b1$_{BL/6}$ binding to ROP5C can account for the residual protein amounts at the PVM in *T. gondii* virulent strain-infected *Irgb2-b1*$_{CIM}$ ko cells complemented with Irgb2-b1$_{BL/6}$ (Fig. 4c).

The possibility that Irgb2-b1$_{BL/6}$ and Irgb2-b1$_{Chimera}$ interaction with ROP5C might contribute to parasite control was investigated by flow cytometry of infected cells. Expression of these particular proteins in *Irgb2-b1*$_{CIM}$ ko cells did not result in increased growth inhibition of virulent *T. gondii* compared to CIM wt cells or *Irgb2-b1*$_{CIM}$ ko cells complemented with Irgb2-b1$_{CIM}$ (Fig. 4d), suggesting that ROP5C does not contribute significantly to virulence of Eurasian *T. gondii* strains in CIM cells.

**T. gondii strains from South America are not restricted by Irgb2-b1$_{CIM}$.** *T. gondii* isolates from South America are genetically highly diverse, and the majority is associated with high mortality rates in laboratory mice. We compared virulence of *T. gondii* strains VAND and AS28, both originating from South America, in CIM mice. Unlike virulent Eurasian *T. gondii* strains[21], VAND (Fig. 5a) and AS28 (Fig. 5b) are lethal in CIM mice.

**T. gondii VAND-derived ROP5 isoforms are not targeted by Irgb2-b1$_{CIM}$.** The first step to identify the molecular constituents that account for virulence of *T. gondii* strains from South America in CIM mice was to determine activity of the ROP5/ROP18/GRA7 kinase complex on Irga6$_{CIM}$. We assessed Irga6$_{CIM}$ phosphorylation in *T. gondii*-infected CIM cells by immunofluorescence analysis. Mean p(T108)Irga6$_{CIM}$ intensities were increased in CIM cells infected with *T. gondii* VAND, but not with AS28, compared with RHΔ*hxgprt* (Fig. 6a). Since high p(T108)Irga6 levels have already been shown to correlate with low

Irgb2-b1 intensities[21], we considered the possibility that Irgb2-b1$_{CIM}$ levels on VAND-derived vacuoles in CIM cells might be lower than on RHΔ*hxgprt*. This, in turn, would suggest that Irgb2-b1$_{CIM}$ interacts inefficiently with ROP5 variants expressed by VAND. We therefore determined vacuolar Irgb2-b1$_{CIM}$ intensities in *T. gondii* RHΔ*hxgprt*-, VAND- and AS28-infected CIM cells. Irgb2-b1$_{CIM}$ intensities at vacuoles of both AS28 and VAND were clearly decreased relative to RHΔ*hxgprt* (Fig. 6b, representative images are shown in Supplementary Fig. 8). Indeed, in the case of VAND, Irgb2-b1$_{CIM}$ levels were similar to those observed at vacuoles of RHΔ*rop5* parasites[21], suggesting that VAND ROP5 does not interact at all with Irgb2-b1$_{CIM.}$ These results were confirmed in pull-down experiments from tachyzoite lysates with GST-tagged Irgb2-b1$_{CIM}$; binding of Irgb2-b1$_{CIM}$ to ROP5 was reduced in case of *T. gondii* AS28 and not detectable in case of *T. gondii* VAND (Fig. 6c). Input of GST-Irgb2-b1$_{CIM}$ (Fig. 6c, upper left hand panels) and protein amounts in the tachyzoite lysates used in the pull-down (Fig. 6c, right hand panels) is shown. These results correlate well with results obtained by immunofluorescence analysis (Fig. 6b). Equal ROP5 expression levels and recognition of AS28 and VAND ROP5 variants by the anti-ROP5 antibody used for western blot analysis could be confirmed after immunoprecipitation from extracellular tachyzoite lysates (Supplementary Fig. 9). Recently we demonstrated that *T. gondii* GRA7 is associated with ROP5[43] but surprisingly, VAND GRA7 was detected in the GST-Irgb2-b1$_{CIM}$ pull-down in the absence of ROP5 binding (Fig. 6c). This proved to be due to a direct interaction between VAND GRA7 and empty GST beads (Supplementary Fig. 10a). Further evidence that GRA7 is not a functional inhibitor of VAND ROP5 association with Irgb2-b1$_{CIM}$ was provided by Y2H analysis. Here, in the absence of VAND GRA7, none of the VAND ROP5 isoforms, ROP5A, B1, B2 and B3, interacted with Irgb2-b1$_{CIM}$ (Fig. 6d), suggesting that an intrinsic polymorphism of ROP5$_{VAND}$ is responsible for its failure to interact with Irgb2-b1$_{CIM}$. Moreover, the YTH results confirm that absolute absence of ROP5$_{VAND}$ binding to Irgb2-b1 after pull-down (Fig. 6c) is not simply due to failure of the antibody used in the western blot to recognize ROP5$_{VAND}$ isoforms. ROP5 uses a surface for binding to Irga6$_{BL/6}$ that is highly polymorphic and under positive selection[49]. When *T. gondii* RH and VAND *rop5* alleles are compared, several amino acid substitutions within this region can be found[41,50] (Fig. 6e). We created a ROP5B3$_{VAND}$ mutant carrying the ROP5B$_{RH}$ interface for Irga6$_{BL/6}$ binding (ROP5B$_{Chimera}$) and investigated binding to Irgb2-b1$_{CIM}$. In the PCA assay, binding of ROP5B3$_{VAND}$ to Irgb2-b1$_{CIM}$ is significantly reduced compared to ROP5B$_{Chimera}$ and Irgb2-b1$_{CIM}$ (Supplementary Fig. 11).

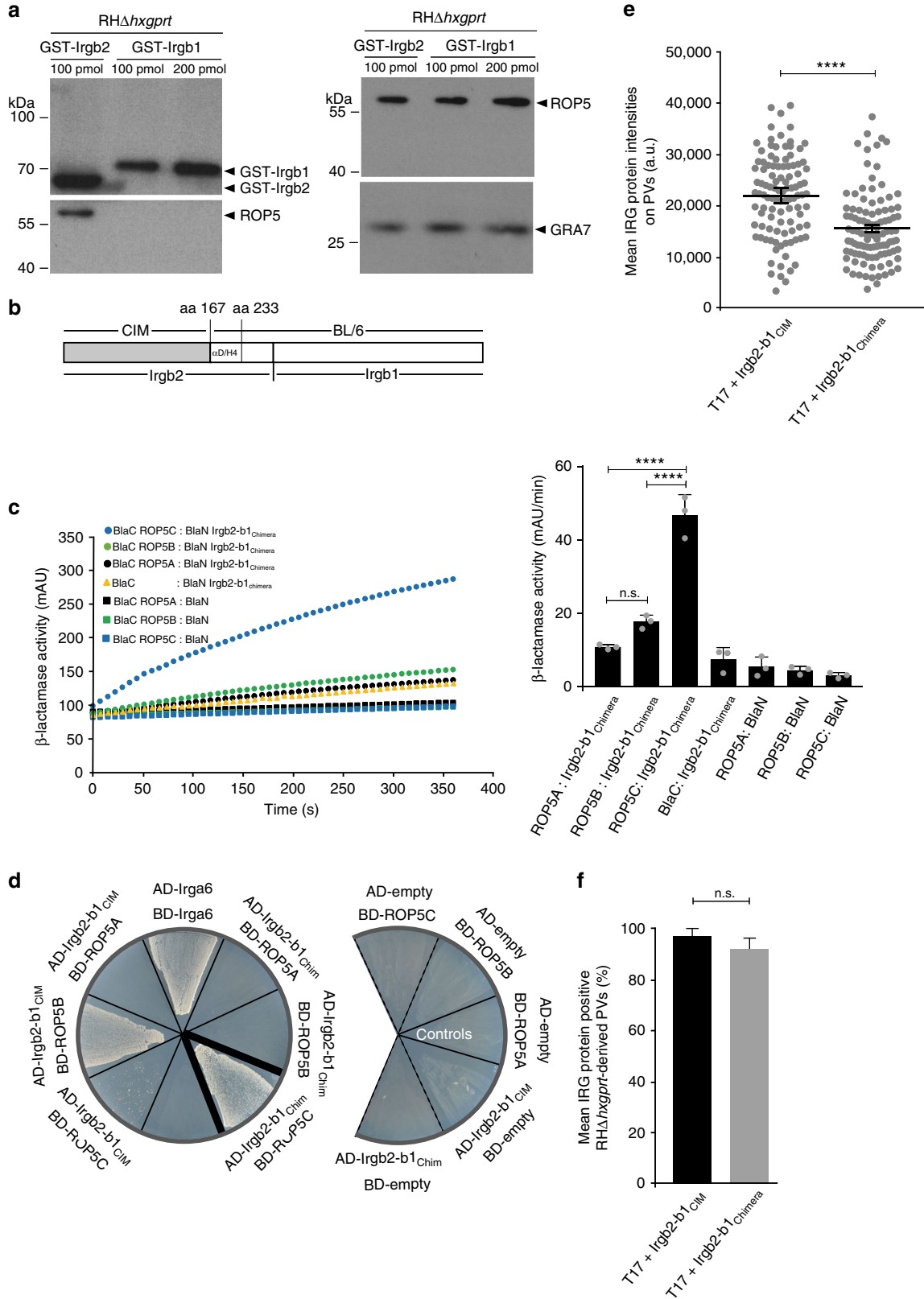

## Discussion

Restriction of *T. gondii* growth in mice is dependent on IFNγ-inducible IRG and GBP (guanylate-binding protein) proteins that accumulate at the PVM leading to its rupture, death of the parasite and necrotic death of the host cell[16,51]. *T. gondii* virulent strains are able to overcome cell-autonomous resistance by secretion of effector proteins that inactivate IRG and GBP protein function[52,53]. Certain *T. gondii* strains of the RFLP genotype 10 (formerly type I) are highly virulent for laboratory mice but are well resisted in CIM wild-derived mice. In the latter, polymorphic

**Fig. 3** The Irgb2-b1$_{CIM}$ interface for *T. gondii* ROP5B binding is polymorphic. **a** Pull-down of *T. gondii* ROP5 by GST-Irgb2$_{CIM}$ but not GST-Irgb1$_{CIM}$ from RHΔ*hxgprt* tachyzoite detergent lysates (lower left hand panel). The upper left hand panel indicates input of glutathione *S*-transferase (GST)-fusion proteins in the pull-down. The right hand panels display amounts of ROP5 (upper panel) and GRA7 (lower panel) in the tachyzoite lysate used for the pull-down. **b** Schematic representation of Irgb2-b1 CIM-BL/6 chimeric variant (Irgb2-b1$_{Chimera}$) used in (**c, d**). **c, d** Irgb2-b1$_{CIM}$ interaction with *T. gondii* ROP5B but not ROP5C is disturbed in case of Irgb2-b1$_{Chimera}$. **c** Protein-fragment complementation assay. Proteins were fused to N-terminal (BlaN) or C-terminal (BlaC) fragments of the reporter protein *TEM-1* β-lactamase. Error bars indicate the mean and standard deviation of three independent experiments (right hand panel). One-way analysis of variance (ANOVA) followed by Tukey's multiple comparison was used to test differences between groups; ****$p < 0.0001$; n.s. not significant. The kinetic of the β-lactamase reaction is shown for one representative experiment (left hand panel). Interaction of Irgb2-b1$_{CIM}$ with *T. gondii* ROP5B is almost completely abrogated with Irgb2-b1$_{Chimera}$. **d** Yeast two-hybrid approach. Proteins were expressed either as fusion to a transcriptional activation domain (AD) from pGAD-C3 or to a DNA-binding domain (BD) from pGBD-C3. Colony growth under 3DO conditions is indicative of protein/protein interaction. Black dotted lines indicate assembly of relevant areas from two different plates. **e** Intensities of individual vacuoles positive for Irgb2-b1$_{CIM}$ and Irgb2-b1$_{Chimera}$ detected by immunofluorescence with anti-Irgb2-b1-specific antiserum in interferon-γ (IFNγ)-induced (200 U ml$^{-1}$) wild-type (wt) CIM diaphragm-derived cells (DDCs) infected for 2 h with RHΔ*hxgprt*. Intensities of Irgb2-b1$_{Chimera}$ are significantly reduced at RHΔ*hxgprt*-derived vacuoles. **f** Frequency of Irgb2-b1$_{CIM}$- and Irgb2-b1$_{Chimera}$-positive vacuoles detected by immunofluorescence with anti-Irgb2-b1-specific antiserum in IFNγ-induced (200 U ml$^{-1}$) complemented *Irgb2-b1* ko CIM DDCs infected for 2 h with RHΔ*hxgprt* (about 100 vacuoles were counted per experiment). No differences in numbers of Irgb2-b1$_{CIM}$- and Irgb2-b1$_{Chimera}$-positive vacuoles could be observed. In **e** and **f**, error bars indicate the mean and SEM of three independent experiments. Student's t-test was used for two-group comparisons; ****$p < 0.0001$, n.s. not significant

IRG proteins on Chr11 were shown to protect CIM mice from *T. gondii* virulent strain infections and the highly polymorphic tandem IRG protein Irgb2-b1$_{CIM}$ was implicated[21]. In this study, we have shown by gene deletion and complementation that polymorphic variation in Irgb2-b1 is indeed responsible for the restriction of virulent *T. gondii* strains in CIM cells and the failure of restriction in C57BL/6 (BL/6) cells (Fig. 7). Two independent *Irgb2-b1*$_{CIM}$ ko CIM cell lines lost the ability to control *T. gondii* virulent strain replication compared to wt cells. Complementation of ko cells with *Irgb2-b1*$_{CIM}$ restored resistance to wt levels (Fig. 1e). Phosphorylation of Irga6 is a good indicator for parasite virulence[39,40] and loss of parasite control in the absence of *Irgb2-b1*$_{CIM}$ is reflected in elevated numbers of p(T108)Irga6$_{CIM}$-positive vacuoles upon virulent strain infection (Fig. 1c).

In our earlier study, overexpression of Irgb2-b1$_{CIM}$ in transiently transfected BL/6 cells rescued endogenous Irga6$_{BL/6}$ from ROP5/ROP18/GRA7-mediated phosphorylation. In the same study Irgb2-b1$_{CIM}$ was itself shown to be phosphorylated by virulent ROP5/ROP18 but this phosphorylation did not inhibit its PVM accumulation. Surprisingly and in stark contrast to other IRG proteins, vacuolar accumulation of Irgb2-b1$_{CIM}$ is strictly dependent on *T. gondii*-derived ROP5; lower ROP5 levels correlate with decreased amounts of Irgb2-b1$_{CIM}$[21]. Based upon these results, we investigated binding of Irgb2-b1$_{CIM}$ to ROP5 as the underlying molecular mechanism of CIM-inherent resistance. Here we demonstrate direct binding of Irgb2-b1$_{CIM}$ to *T. gondii*-derived ROP5. The result provides corroborative evidence that Irgb2-b1$_{CIM}$ traps ROP5, leading to diversion of the ROP18 kinase function from Irga6 to Irgb2-b1, and in that way confers cell-autonomous resistance against virulent *T. gondii* strains. Interestingly, binding of Irgb2-b1$_{CIM}$ to ROP5 is isoform specific. The impact of ROP5 isoforms on *T. gondii* virulence has been investigated in former studies. Single or pair-wise combinations of *rop5* isoforms were tested to rescue the virulence phenotype in RHΔ*rop5* organisms. Complementation with one or two copies of *rop5*A$_{RH}$ only partially rescued virulence and only *rop5*A$_{RH}$ in combination with *rop5*B$_{RH}$ displayed a phenotype indistinguishable from parental strain infections[49]. Otherwise, when ROP5A$_{RH}$ or ROP5C$_{RH}$ was expressed alone in a *T. gondii* avirulent genetic background, Irgb6$_{BL/6}$ loading was decreased compared to wt strain infections. However, none of these ROP5 isoforms increased parasite virulence in mice[41]. The effect of ROP5B$_{RH}$ alone on *T. gondii* virulence has never been investigated. In all our assays, efficient binding of Irgb2-b1$_{CIM}$ could be observed only to ROP5B$_{RH}$ but not to ROP5C$_{RH}$ or ROP5A$_{RH}$. Irgb2-b1$_{BL/6}$ was strongly associated with ROP5C$_{RH}$; the meaning of this

unexpected result is not immediately clear since Irgb2-b1$_{BL/6}$ is expressed at very low levels in BL/6 cells. Furthermore, complementation of an *Irgb2-b1*$_{CIM}$ ko cell line with Irgb2-b1$_{BL/6}$ does not confer resistance against *T. gondii* virulent strain infection. Certainly these findings indicate a predominant responsibility of ROP5B$_{RH}$ as opposed to the A or C isoforms for the virulence of virulent Eurasian *T. gondii* strains in laboratory mouse strains, and Irgb2-b1$_{CIM}$ is its principal antagonist in CIM mice and presumably other strains that have the same *Irgb2-b1* allele. Low expression levels of endogenous Irgb2-b1$_{BL/6}$ result in negligible numbers of Irgb2-b1$_{BL/6}$-positive vacuoles of virulent *T. gondii* and correspondingly high virulence. However, the failure of Irgb2-b1$_{BL/6}$, unlike Irgb2-b1$_{CIM}$, to attenuate virulence is due not only to its low expression level but also to its different amino acid sequence since overexpression of Irgb2-b1$_{BL/6}$ in BL/6 cells increases the number of Irgb2-b1$_{BL/6}$-positive RHΔ*hxgprt* strain vacuoles (presumably this Irgb2-b1$_{BL/6}$ association with the PVM is due to binding to ROP5C (Fig. 4)) but does not increase endogenous Irgb6$_{BL/6}$ loading[21]. Comparative sequence analysis suggested that so far the N-terminal unit, Irgb2, of the Irgb2-b1 tandem is the only *IRG* sequence that is under recent divergent selection[21], suggesting that this may be the unit that interacts directly with a cognate ROP5 isoform. Indeed, we were able to demonstrate binding of polymorphic Irgb2$_{CIM}$ but not Irgb1$_{CIM}$ to *T. gondii* virulent strain-derived ROP5B (Fig. 5). Two hundred nucleotides in *Irgb2*$_{CIM}$ (nt500–nt700) represent a significant hotspot of divergent selection among mouse strains encompassing structural motifs αd and H4. Both αd and H4 constitute essential parts of the polymorphic Irga6$_{BL/6}$ interface involved in ROP5 binding[38,47] and we could show that these structural elements are also involved in the binding of Irgb2-b1$_{CIM}$ to ROP5B but not ROP5C. Like Irgb2-b1$_{BL/6}$, the Irgb2-b1$_{Chimera}$ associated strongly with ROP5C and an *Irgb2-b1*$_{CIM}$ ko cell line complemented with Irgb2-b1$_{Chimera}$ did not confer resistance against *T. gondii* infection, again indicating a minor role for ROP5C in these *T. gondii* and mouse strain combinations.

*T. gondii* isolates from South America are genetically highly diverse[4,10–14] and the majority is associated with high mortality rates in laboratory mice[11,15], and in some cases ROP5 and ROP18 could be demonstrated to be major virulence determinants[50]. Interestingly, all *T. gondii* strains from South America tested so far are lethal even in CIM mice (Fig. 5, U. B. Müller and J. C. Howard, unpublished results). To examine the molecular constituents that account for virulence of *T. gondii* strains from South America in CIM mice, we determined levels of p(T108)Irga6$_{CIM}$ and Irgb2-b1$_{CIM}$ at the PVM. In VAND-infected IFNγ-induced

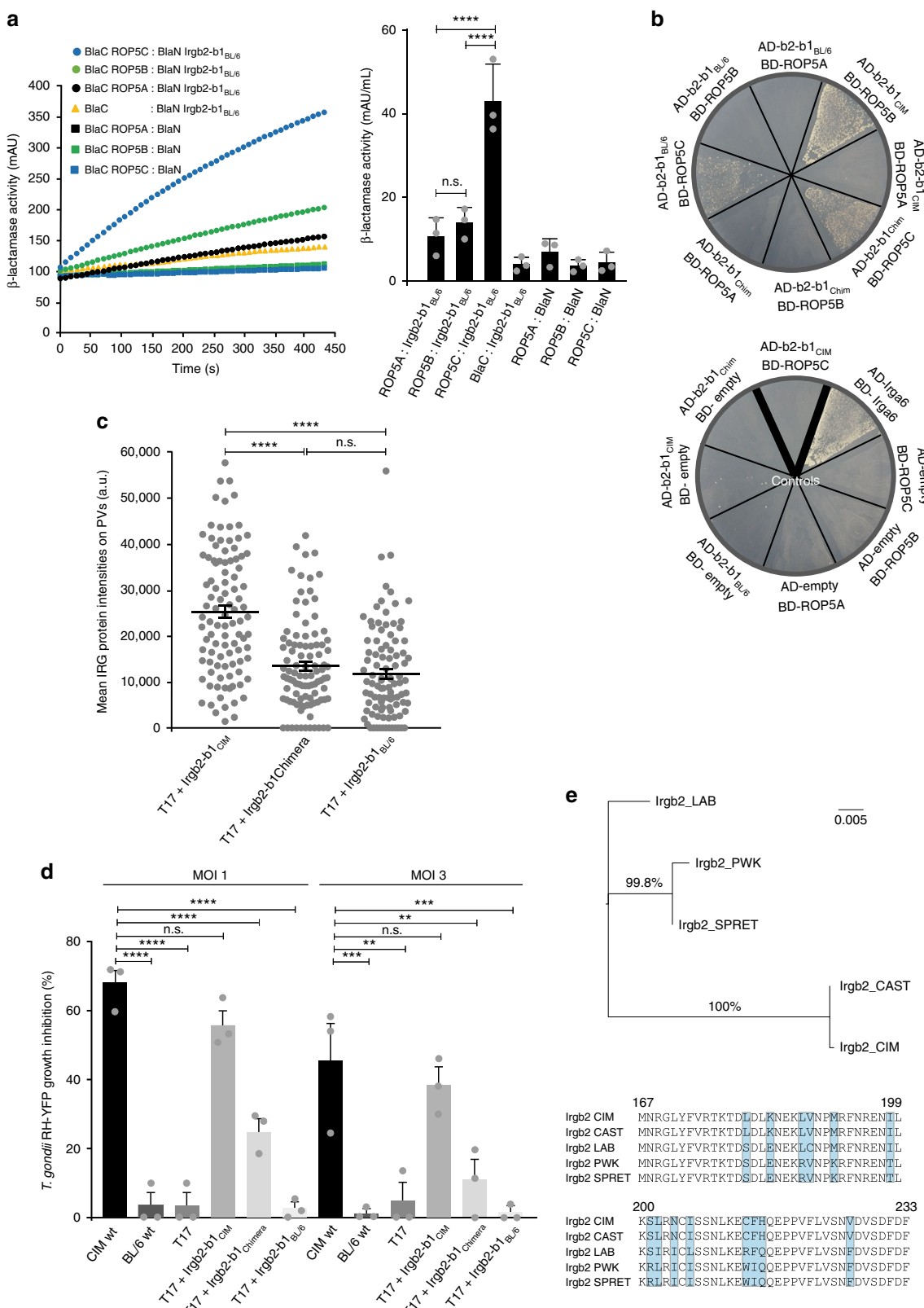

CIM cells, the absence of Irgb2-b1$_{CIM}$ on the vacuole was associated with elevated levels of p(T108)Irga6$_{CIM}$. In case of AS28, the Irgb2-b1$_{CIM}$ loading phenotype was intermediate between RHΔ*hxgprt* and VAND, but perhaps sufficient to account for the complete absence of Irga6$_{CIM}$ phosphorylation (Fig. 6a, b). Consistently, limited binding of AS28-derived ROP5 to Irgb2-

b1$_{CIM}$ could be detected, whereas no interaction with Irgb2-b1$_{CIM}$ for any of the VAND ROP5 isoforms was visible in our assays (Fig. 6). In summary, in VAND infections, escape from Irgb2-b1$_{CIM}$ binding conclusively results in increased Irga6$_{CIM}$ phosphorylation and is therefore a very likely explanation for virulence of *T. gondii* VAND in CIM cells and mice (Fig. 7).

**Fig. 4** Irgb2-b1_BL/6 directly binds ROP5C. **a** Protein-fragment complementation assay. Proteins were fused to N-terminal (BlaN) or C-terminal (BlaC) fragments of *TEM-1* β-lactamase. One-way analysis of variance (ANOVA) followed by Tukey's multiple comparison was used to test differences between groups; ****$p < 0.0001$; n.s. not significant. The kinetic of the reaction is shown for one representative experiment (left hand panel). Irgb2-b1_BL/6 interacts with *T. gondii* ROP5C but not ROP5B and ROP5A. **b** Yeast two-hybrid. Proteins were expressed as fusion with transcriptional activation domain (AD) from pGAD-C3 or DNA-binding domain (BD) from pGBD-C3. Colony growth is indicative of Irgb2-b1_BL/6 and Irgb2-b1_Chimera interaction with ROP5C and Irgb2-b1_CIM interaction with ROP5B. Bold black lines separate samples from negative controls. **c** Intensities of individual vacuoles detected by immunofluorescence with anti-Irgb2-b1-specific antiserum in interferon-γ (IFNγ)-induced (200 U ml$^{-1}$) cells infected for 2 h with RHΔ*hxgprt*. Kruskal–Wallis test followed by Dunn's multiple comparisons was used to test differences between groups; ****$p < 0.0001$; n.s. not significant. Intensities of Irgb2-b1_BL/6 and Irgb2-b1_Chimera are reduced at RHΔ*hxgprt*-derived vacuoles. **d** Polymorphic Irgb2-b1_CIM interface mediating ROP5B binding is crucial for *T. gondii* control in vitro. CIM diaphragm-derived cells (DDCs) were induced with 100 U ml$^{-1}$ IFNγ for 24 h and infected with *T. gondii* RH-YFP at a multiplicity of infection (MOI) of 1 or 3. Parasite growth was determined by flow cytometry 24 h post infection. One-way ANOVA followed by Dunnett's multiple comparisons was used to test differences between CIM wild-type (wt) and other cells; ****$p < 0.0001$; ***$p < 0.0008$; **$p < 0.006$; n.s. not significant. Control of *T. gondii* virulent strain replication is lost in Irgb2-b1_CIM ko CIM DDCs complemented with *Irgb2-b1*_Chimera. **e** Upper panel, phylogenetic analysis and maximum likelihood tree of *Irgb2* sequences (CIM, *M. m. castaneus*; CAST, *M. m. castaneus*; LAB, *M. m. domesticus*; PWK, *M. m. musculus*; SPRET, *Mus spretus*). LAB indicates the allele shared by all laboratory mice tested so far, including C57BL/6[21]. Only bootstrap values above 50 are shown. Lower panel, alignment of Irgb2 amino acids M167 to P233 encompassing putative structural motifs αd and H4. Polymorphic sites are highlighted in blue. Error bars indicate the mean and SEM or standard deviation of three independent experiments

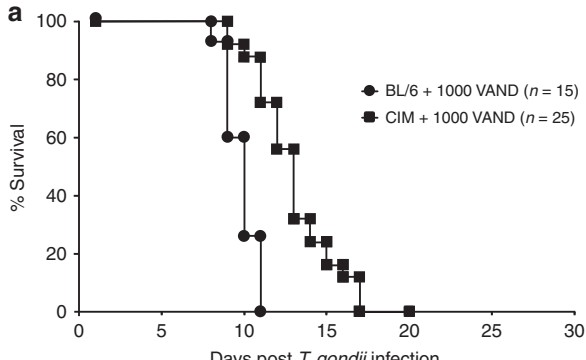

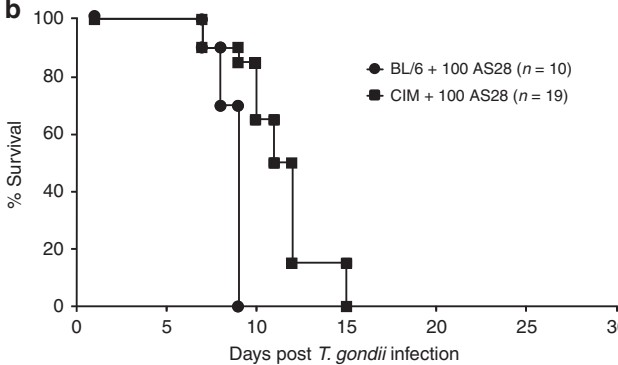

**Fig. 5** South American *T. gondii* VAND and AS28 evade growth restriction in CIM mice. CIM wild-type (wt) (**a**, $n = 25$; **b**, $n = 19$) and C57BL/6 (**a**, $n = 15$; **b**, $n = 10$) mice were infected by intraperitoneal injection of 1000 *T. gondii* VAND (**a**) or 100 AS28 (**b**) strain parasites and survival monitored for 15–20 days. Data shown are combined from three (**a**) or two (**b**) independent experiments

Decreased vacuolar Irgb2-b1_CIM levels in *T. gondii* AS28 infections, on the other hand, do not result in increased Irga6_CIM phosphorylation, implicating either different molecular virulence mechanisms not associated with the IRG system or possibly different effector IRG protein targets, not including Irga6.

It would appear that the allelic diversity of the tandem IRG protein Irgb2-b1 present in laboratory or some wild-derived strains such as CIM does not include a sequence capable of fully inhibiting the ROP5/ROP18/GRA7 virulence kinase complex of VAND and probably other South American strains. It is

interesting to consider whether indeed a suitable *Irgb2-b1* allele exists anywhere in the mouse species, and if not, what other, presumably South American, intermediate host species are capable of attenuating the extreme virulence of these *T. gondii* strains. For the time being, it seems legitimate to consider that the ROP5/ROP18/GRA7 virulence kinase complex is specifically directed against IRG effector proteins. Everything points to accurate molecular complementarity as the basis for the function of the kinase complex, both at the level of the ROP5/IRG interface[38,41,43,47] and the accurate targeting of the rhoptry kinase to the sensitive threonines in the IRG switch region[39,40,43]. Although GBP have been demonstrated to be associated with resistance to *T. gondii* in mice[35,52], there is presently no indication that alleles of the ROP5/ROP18/GRA7 system can also be deployed to directly target and phosphorylate this entirely different protein family. Indeed, there is already some evidence that the striking differential virulence of *T. gondii* strains in mice is not reflected in comparable differences of virulence in humans, where the IRG system has been lost[41]. Lethality of *T. gondii* for healthy humans is exceedingly rare, and the main differential pathologies seem to be associated primarily with damage to the optic system[54,55]. A reasonable hypothesis is that the ROP5/ROP18/GRA7 kinase system has specifically evolved to attenuate an attack by effector proteins of the IRG system on early stages of tachyzoite expansion. The opposed polymorphisms connecting virulence and resistance suggest complex evolutionary dynamics beyond the species level and on a global scale. While it is evident that a big part of the resistance component is due to IRG proteins, it is unlikely that IRG proteins of *Mus musculus*, an Old World species, are alone responsible for the binding specificities of ROP5 evolved in *T. gondii*, a parasite almost certainly of New World origin. It is more likely that *Irgb2-b1* alleles of evolutionarily significant New World mammals will turn out to be the key resistance factors against New World *T. gondii* strains.

## Methods

**Propagation of *T. gondii*.** Tachyzoites of *T. gondii* strains RHΔ*hxgprt*[56], RH-YFP[57], ME49[58], RHΔ*rop5*[37], VAND[59] and AS28[60] were cultivated in confluent monolayers of human foreskin fibroblasts (HS27, ATCC CRL-1634), harvested and immediately used for infection of cells or lysed for subsequent immunoprecipitation or pull-down experiments.

**Cell culture.** HEK293T cells (ATTC; CRL-3216) and wt or ko diaphragm-derived cells (DDCs) derived from C57BL/6 and CIM mice[21] were maintained by serial passage in Dulbecco's modified Eagle's medium, high glucose (Invitrogen Life Technologies) supplemented with 2 mM L-glutamine, 1 mM sodium pyruvate, 1× Minimum Essential Medium non-essential amino acids, 100 U ml$^{-1}$ penicillin, 100 mg ml$^{-1}$ streptomycin (PAA) and 10% fetal calf serum (FCS, Biochrom).

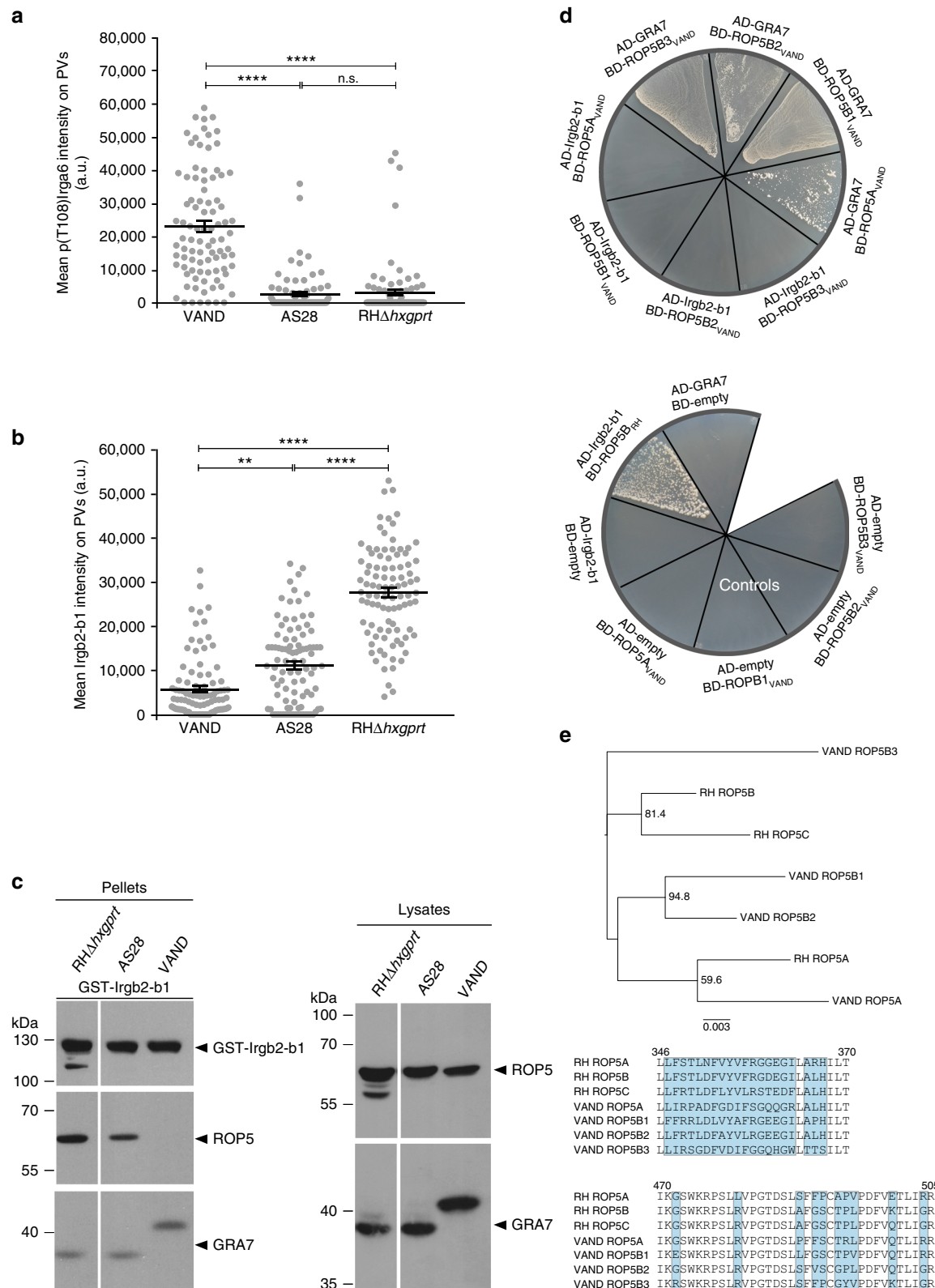

Human foreskin fibroblasts (HS27, ATCC; CRL-1634) were maintained in Iscove's modified Dulbecco's medium, high glucose (Invitrogen Life Technologies) supplemented with 100 U ml$^{-1}$ penicillin, 100 mg ml$^{-1}$ streptomycin and 5% FCS. All cells were mycoplasma-free and regularly tested by PCR[61].

**Immunological reagents**. Immunoreagents used in this study were: 3E2 mouse monoclonal antibody against ROP5 isoforms[62], affinity-purified rabbit sera 87558 against (pT108)Irga6 (1:8000)[39], 10E7 mouse monoclonal antibody (1:2000)[63] against Irga6, B34 mouse monoclonal antibody (1:2000)[64] against Irgb6, 940/6 rabbit antiserum (1:2000)[43] against Irgb10, 2078 rabbit antiserum (1:1000)[65] against Irgd, 954/1-C15A rabbit antiserum (1:8000 in western blot; 1:4000 in immunofluorescence)[21] against a conserved Irgb-tandem C-terminal peptide, 3.1.2 (1:500) and 2.4.21 (1:1000) rat monoclonal antibodies against *T. gondii* GRA7[43], anti-GST goat antiserum (1:1000 GE Healthcare 27457701), anti-FLAG mouse monoclonal antibody (1:1000, Sigma Aldrich F3165), mouse monoclonal anti-actin

**Fig. 6** *T. gondii* VAND-derived ROP5 escapes targeting by Irgb2-b1$_{CIM}$. **a** p(T108)Irga6$_{CIM}$ protein intensities are increased at VAND-derived vacuoles. interferon-γ (IFNγ)-induced CIM diaphragm-derived cells (DDCs; 200 U ml$^{-1}$) were infected for 2 h with indicated *T. gondii* strains and individual p(T108) Irga6$_{CIM}$-positive vacuoles identified with 558 p(T108)Irga6$_{CIM}$-specific antiserum. Error bars indicate the mean and SEM of three independent experiments. Kruskal–Wallis test followed by Dunn's multiple comparisons was used to test differences between groups; ****$p < 0.0001$; n.s. not significant. **b** Irgb2-b1$_{CIM}$ intensities are reduced at VAND- and AS28-derived vacuoles. IFNγ-induced CIM DDCs (200 U ml$^{-1}$) were infected for 2 h with indicated *T. gondii* strains and individual Irgb2-b1$_{CIM}$-positive vacuoles identified with anti-Irgb2-b1-specific antiserum. Error bars indicate the mean and SEM of three independent experiments. Kruskal–Wallis test followed by Dunn's multiple comparisons was used to test differences between groups; ****$p < 0.0001$; **$p < 0.006$. **c** In vitro pull-down with recombinant GST-Irgb2-b1$_{CIM}$ fusion protein and *T. gondii* tachyzoite detergent lysates. Pull-down of ROP5 by Irgb2-b1$_{CIM}$ is reduced with AS28 and completely lost in case of VAND tachyzoite lysates compared to RHΔ*hxgprt*-derived ROP5 (middle left hand panel). VAND GRA7 pull-down is not dependent on ROP5 (lower panel). The upper panel indicates input of GST-Irgb2-b1$_{CIM}$ in the pull-down. The right hand blot shows ROP5 (upper panel) and GRA7 (lower panel) levels in tachyzoite lysates. All tracks were run on a single gel; vertical white lines indicate excision of irrelevant tracks. **d** VAND ROP5 isoforms ROP5A, ROP5B1, ROP5B2 and ROP5B3 do not directly interact with Irgb2-b1$_{CIM}$ or VAND GRA7 in a yeast two-hybrid approach. Proteins were expressed either as fusion to a transcriptional activation domain (AD) from pGAD-C3 or to a DNA-binding domain (BD) from pGBD-C3. Colony growth under 3DO conditions is indicative of protein/protein interaction. Bold black line separates samples from negative controls. **e** Upper panel, alignment of *T. gondii* RH and VAND ROP5 amino acid sequences that represent a polymorphic hotspot and have been shown to be involved in binding to Irga6$_{BL/6}$. Polymorphic sites are highlighted in blue. Lower panel, phylogenetic analysis and maximum likelihood tree of *T. gondii* RH and VAND *rop5* sequences. Only maximum likelihood bootstrap values above 50 are shown

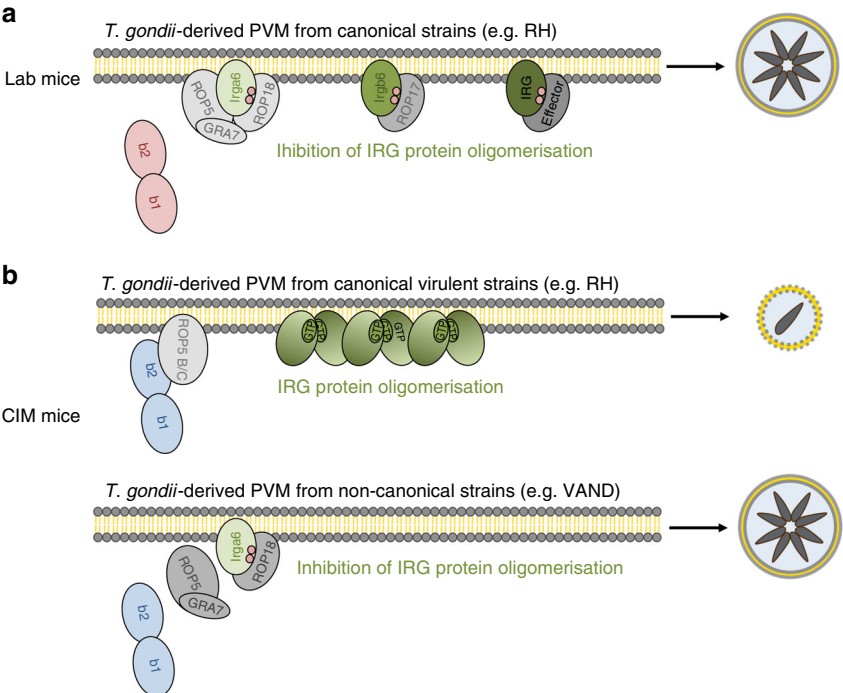

**Fig. 7** Model for Irgb2-b1-mediated control of *T. gondii* infection in wild-derived mice. **a** In laboratory mice, effector proteins from canonical virulent *T. gondii* strains (like RH) specifically phosphorylate certain Immunity-Related GTPases (IRG) proteins, thereby inhibiting oligomerisation and destruction of the parasitophorous vacuolar membrane (PVM). Two rhoptry kinases, ROP18 and ROP17, have been demonstrated to preferentially phosphorylate Irga6$_{BL/6}$ or Irgb6$_{BL/6}$. The existence of additional parasite effectors specific for other IRG proteins is assumed. **b** In CIM and probably other wild-derived mice, canonical virulent *T. gondii* strains (like RH) are counteracted by direct binding of ROP5 isoform B by the polymorphic tandem IRG protein Irgb2-b1$_{CIM}$. IRG effector proteins are free to accumulate around the PVM, resulting in growth control, encystment and transmission of the parasite. Genetically more diverse *T. gondii* strains, like *T. gondii* VAND from South America, express polymorphic ROP5 variants that are not targeted by Irgb2-b1$_{CIM}$. Consequently, effector IRG proteins such as Irga6$_{CIM}$ are phosphorylated and inactivated by a *T. gondii* VAND kinase complex. Infected animals die shortly after parasite challenge. Molecular interaction between ROP18 and ROP5 or Irga6 and ROP5 within the VAND kinase complex awaits experimental confirmation

antibody (1:1000, Sigma Aldrich A3853) and anti-calnexin rabbit antiserum (1:1000, Merck 208880).

Alexa 488 and Alexa 555 labeled donkey anti-rat (A21208) and donkey anti-rabbit (A31572) fluorescent antisera (1:1000, Thermo Fisher Scientific), goat anti-rabbit-HRP (111-035-045), goat anti-rat-HRP (112-035-003), donkey anti-goat-HRP (705-035-147) and rabbit anti-mouse-HRP (315-035-045) (all 1:5000, Jackson Immuno Research Laboratories) polyclonal antibodies were used as secondary reagents.

**Lysate preparation from free tachyzoites and infected cells**. The $10–25 \times 10^6$ free *T. gondii* tachyzoites or $2.5 \times 10^6$ DDCs seeded in 10 cm plates were stimulated with 200 U ml$^{-1}$ IFNγ for 24 h, subsequently infected for 2 h with *T. gondii* at a multiplicity of infection (MOI) of 10 and washed trifold with phosphate-buffered saline (PBS), and lysed in 800 µl NP-40 lysis buffer (0.1% NP-40, 150 mM NaCl, 20 mM Tris/HCl (pH 7.6), 5 mM MgCl$_2$ supplemented with protease inhibitors (Roche)) for 2 h under constant rotation at 4 °C. Postnuclear lysates were subjected to immunoprecipitation or pull-down analysis.

**Immunoprecipitation and pull-down analysis**. For immunoprecipitation experiments, postnuclear lysates were incubated with the indicated antibodies o/n at 4 °C followed by an additional 1 h of incubation with 100 µl 1:1 (lysis buffer) bead suspension of protein A-Sepharose (Amersham) resin[38]. For pull-down experiments, 100 or 200 pmol (Fig. 3a) of purified GST or GST-fusion proteins were mixed with 100 µl 1:1 bead suspension of glutathione sepharose 4B (GE

Healthcare) resin in 500 µl PBS/2 mM dithiothreitol (DTT) for 1 h at 4 °C. The resin was washed trifold in ice-cold lysis buffer containing 2 mM DTT without detergent and incubated with 800 µl postnuclear *T. gondii* lysate o/n at 4 °C.

Beads were washed trifold with lysis buffer and either stored at −80 °C or immediately boiled in sample buffer (80 mM Tris/HCl (pH 6.8), 5 mM EDTA, 4% SDS, 34% sucrose, 40 mM DTT, 0.002% bromphenol blue) for 5 min at 95 °C and subjected to SDS-PAGE (sodium dodecyl sulfate–polyacrylamide gel electrophoresis) and western blot. Uncropped images of all western blots are provided in the Source Data file.

**T. gondii replication assay.** *T. gondii* proliferation in infected DDCs was determined by incorporation of ³H-uracil[66] or flow cytometry.

For the uracil incorporation assay, cells were grown for 24 h in the presence of IFNγ or left unstimulated, and then infected with *T. gondii* RH-YFP at different MOIs for an additional 24 h. Cultures were subsequently labeled with ³H-uracil (0.3 µCi per well) for 24 h, harvested on glass fiber filters and radioactivity incorporated into proliferating *T. gondii* DNA determined in a beta scintillation spectrometer.

For flow cytometry, $1 \times 10^5$ DDCs seeded in 12-well plates and stimulated with 100 U ml⁻¹ IFNγ for 24 h were infected with RH-YFP. At 24 h post infection, cells were trypsinized, washed 2× with PBS containing 3% FCS (PBS/FCS) and resuspended in 400 µl PBS/FCS containing 1% paraformaldehyde (PFA). After 15 min of incubation at room temperature, fixed cells were washed 2× with PBS/FCS, resuspended in 400 µl PBS/FCS and analyzed by FACSCanto II flow cytometer (BD Biosciences). For each sample, 20,000 events were recorded. Further analysis was performed using FlowJo vX 10.0.7 Software. Percental inhibition of *T. gondii* replication was defined as follows: 100 − (mean IFNγ-stimulated/mean unstimulated) × 100.

**Generation of Irgb2-b1 ko cells.** $2 \times 10^5$ CIM DDCs were seeded in 6-well plates and cells transfected with CRISPR/Cas9 plasmids (see Plasmid constructs) according to the Lipofectamine 3000 protocol (Invitrogen). At 48 h post transfection, 1 cell per 100 µl was seeded into a well of a 96-well plate and each well observed for several days to contain only one single colony. Single colonies were transferred to 24-well plates and detergent lysates of IFN-γ-induced (200 U ml⁻¹) cells 24 h later analyzed by western blot for Irgb2-b1$_{CIM}$ expression.

**Mice virulence assay.** Female and male mice (25 *Mus musculus domesticus*, C57BL/6; 44 *Mus musculus castaneus*, CIM) with ages ranging from 2 to 4 months were infected intraperitoneally with 300 µl or 100 µl of PBS containing freshly harvested tachyzoites of indicated *T. gondii* strains. Survivors were killed at the indicated days post infection and tested for seroconversion using the Toxocell Latex Kit (Biokit). Data shown are combined from independent experiments.

**Expression and purification of GST-fusion proteins.** Recombinant GST-Irgb2-b1$_{BL/6}$, GST-Irgb2-b1$_{CIM}$, GST-Irgb2$_{CIM}$ and GST-Irgb1$_{CIM}$ proteins were expressed from pGEX-4T-2 constructs in *Escherichia coli* BL21 after o/n induction with 0.1 mM IPTG (isopropyl β-D-1-thiogalactopyranosid) at 18 °C. Cells were lysed in PBS/2 mM DTT/protease inhibitor (Complete Mini EDTA-free, Roche) using a microfluidiser (EmulsiFlex-C5, Avestin), lysates cleared by centrifugation at $50,000 \times g$ for 60 min at 4 °C and loaded on a GSTrap FF Glutathione Sepharose affinity column (GE Healthcare) in PBS/2 mM DTT. Proteins were eluted with 10 mM reduced L-glutathione in PBS/2 mM DTT and the protein containing fractions subjected to size exclusion chromatography (Superdex 75, Superdex 200; GE Healthcare)[38].

**Immunocytochemistry.** CIM wt and ko DDCs grown on coverslips were infected with *T. gondii* strains at MOI of 5 for 2 h, washed with PBS, fixed in PBS/4% PFA for 20 min at room temperature and permeabilized in PBS/0.1% saponin for 10 min at room temperature before immunostaining[39]. Microscopy and image analysis was performed blind on coded slides[26]. Intracellular parasites were identified from the pattern of *T. gondii* GRA7 staining.

**Yeast two-hybrid assay.** *Saccharomyces cerevisiae* strain PJ69-4α was incubated with 1 µg of plasmid DNA (pGAD-C3 or pGBD-C3 containing the indicated genes) in transformation buffer (50% PEG 3350, 0.2 M LiAc, 0.5 mg ml⁻¹ single-stranded DNA, 0.1 M DTT) for 30 min at 42 °C. Cotransformants were selected by plating on double dropout media (SD/-Leu/-Trp). Colonies grown on double dropout media were replica plated again on double dropout media before OD$_{600}$ measurement of single colonies resuspended in liquid triple dropout media (SD/-Leu/-Trp/-His). Same amount of material was plated on triple dropout media containing 1 mM 3-AT and incubated for 5 to 10 days at 30 °C.

**Protein-fragment complementation assay.** The PCA is based on split *TEM-1* β-lactamase (Bla) of *E. coli*[67]. Two fragments of the reporter protein (Bla) were fused to two putative interaction partners. The individual Bla fragments are non-functional unless proximity upon interaction of the fused proteins of interest is restored. $7.5 \times 10^5$ HEK293T cells seeded in 6-well plates were co-transfected with

1 µg respective plasmid DNA using Lipofectamine 3000 reagent following the manufacturer´s instructions (Invitrogen). At 24 h post transfection, cells were trypsinized, washed 1× with PBS and resuspended in 150 µl passive lysis buffer (Promega) containing protease inhibitor cocktail (Roche). After 45 min of incubation on ice and centrifugation for 30 min at $15,000 \times g$ and 4 °C, 50 µl of supernatants were mixed with 15 µl nitrocefin (Abcam), 15 µl H₂O and 120 µl PBS in a 96-well plate. The β-lactamase-mediated hydrolysis of nitrocefin was measured by the change of absorbance at 495 nm at intervals of 8–9 s for 50 cycles. In the presence of a standard substrate concentration, the actual nitrocefin hydrolysis rate is dependent on the amounts of reconstituted Bla, consequently on the interaction between the fusion proteins. Therefore, to determine the strength of the interaction, the nitrocefin hydrolysis rates, expressed in mAU min⁻¹, were calculated for the linear phase of the reaction and compared to each other and the background rates, which were observed upon transfection of the respective fusion proteins alone[67,68]. All PCA assays were carried out three times and the differences between average hydrolysis rates were compared to evaluate the strength of the interactions.

**Plasmid constructs.** The pGEX-4T-2-Irgb2-b1, pGEX-4T-2-Irgb2 and pGEX-4T-2-Irgb1 constructs allowing expression of recombinant GST-Irgb2-b1, GST-Irgb2 or GST-Irgb1 protein were generated from pGW1H-Irgb2-b1$_{CIM}$ and pGW1H-Irgb2-b1$_{BL/6}$[21]. Irgb2-b1$_{CIM}$, Irgb2-b1$_{BL/6}$ and Irgb2-b1$_{Chimera}$ were amplified from pGEX-4T-2-Irgb2-b1$_{CIM}$ and pGEX-4T-2-Irgb2-b1$_{BL/6}$ and ligated into pGAD-C3, pGBD-C3 and BlaN/BlaC[68,69]. The complete coding sequences of VAND *rop5* isoforms[41] were subcloned from IDT (Integrated DNA Technologies) vectors into pGAD-C3 and pGBD-C3. VAND *gra7* was amplified from *T. gondii* strain VAND genomic DNA and ligated into pGAD-C3 or pGBD-C3.

RH *rop5* isoforms were amplified from respective plasmids[43] and ligated into pGAD-C3, pGBD-C3 and BlaN/BlaC. A complete list of primers is given in Supplementary Table 1.

The pGBD-C3-ROP18 construct was generated earlier[39].

The following gRNAs were hybridized with their complementary strand and cloned into pX335 (Addgene) at the CACCG insertion site to generate *Irgb2-b1*$_{CIM}$ ko constructs:

Irgb2-b1 1: 5'-caccgTGGGTATGATTTTTTCTCAA-3'
Irgb2-b1 2: 5'-caccgGTTGTATATACCACCCCAAC-3'.

**Lentiviral transduction.** Lentiviral transduction was applied to generate *Irgb2-b1*$_{CIM}$ ko cells overexpressing Irgb2-b1 variants. For this purpose, *gag-pol*-expressing and *env*-expressing plasmids were co-transfected with the plasmid carrying the gene of interest into HEK293T cells that have been grow to a density of 70% in a 10 cm plate. At 24 h post transfection, the medium was exchanged and cells incubated for additional 24 h. The supernatant was filtered and transferred to *Irgb2-b1*$_{CIM}$ ko cells (T17) that have been seeded 1 day before in a 6 cm plate and grown to a density of 70%. After 24 h, cells were harvested and transferred into appropriate cells culture flasks with medium containing 1–5 µg ml⁻¹ puromycin for selection of transduced cells.

**Statistics.** All statistical analyses were performed using GraphPad Prism 7.0 software (GraphPad). *P* values were determined by an appropriate statistical test. Statistical differences in IRG protein intensities between groups at single *T. gondii*-derived intracellular vacuoles were determined using a two-tailed Student's *t*-test. One-way analysis of variance (ANOVA) followed by Tukey's multiple comparisons or Kruskal–Wallis test followed by Dunn's multiple comparisons was used to test differences in IRG protein frequencies or intensities between more than two groups at *T. gondii*-derived intracellular vacuoles respectively. Statistical differences for *T. gondii* replication analyzed by FACS were determined using one-way ANOVA followed by Tukey's or Dunnett's multiple comparisons. In case of PCA assays, one-way ANOVA followed by Tukey's multiple comparison was used to test differences between groups. All error bars indicate the mean and standard error of the mean (SEM) or standard deviation of at least three independent experiments. The *p* values < 0.05 were considered to be significant.

**Ethics statement.** All experiments with mice at the University of Cologne were conducted under the regulations and protocols for animal experimentation in accordance with guidelines of the European Commisson (Directive 2010/63/EU) and approved by the local government authorities (Bezirksregierung Köln, Germany), LANUV Nordrhein-Westfalen Permit No. 44.07.189. Procedures using live animals at Instituto Gulbenkian de Ciência were approved by the Instituto Gulbenkian de Ciência ethical committee and by the national animal welfare authority (DGAV) and were carried out in accordance with national (portaria 1005/92) and European (Directive 56/609/CE) regulations

**Reporting summary.** Further information on experimental design is available in the Nature Research Reporting Summary linked to this article.

## Data availability

The authors declare that all data supporting the findings of this study are available within the article and its Supplementary Information files, or are available from the authors

upon request. Associated raw data for Figs. 2, 3, 4, 6 and Supplementary Figures 4, 5, 6, 7, 9, 10, 11 can be found in the source data file.

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

## Acknowledgements

We are especially thankful for all support from the Institute of Virology under the direction of Hartmut Hengel. We thank Andre Riedl for assistance with PCA experiments. Kerstin Flämig and Claudia Kastenholz provided technical assistance. This work was supported by grants from the Deutsche Forschungsgemeinschaft STE 2348/2-1 to T.S., SCHW 632/17-1 to M.S. and SFB635, 670, 680, SPP1399 to J.C.H. M.M.L. and S.S. received funding (Research Grants–Doctoral Programmes in Germany) from the German Academic Exchange Service (DAAD). C.C., C.A. and J.C.H. were supported by the Fundação Calouste Gulbenkian through a grant from the Instituto Gulbenkian de Ciência and by the research infrastructure Congento, project LISBOA-01-0145-FEDER-022170, co-financed by Lisboa Regional Operational Programme (Lisboa 2020), under the Portugal 2020 Partnership Agreement, through the European Regional Development Fund (ERDF), and Foundation for Science and Technology (Portugal). The funders had no role in study design, data collection and interpretation, or the decision to submit the work for publication.

## Author contributions

T.S., M.M.L. and U.B.M. conceived the study; T.S., M.M.L., U.B.M., I.Z., S.S., P.W., C.C., C.A., S.K.-W., N.L. and Z.R. designed experiments; T.S., M.M.L., U.B.M., I.Z., S.S., P.W., C.C., C.A., S.K.-W. and N.L. performed the experiments; T.S., M.M.L., U.B.M., Z.R., J.C.H. and M.S. evaluated the data; T.S., M.M.L. and U.B.M. wrote the manuscript.

## Additional information

**Competing interests:** The authors declare no competing interests.

