## [Peer Review File · Nature Communications]

Reviewers' comments:

Reviewer #1 (Remarks to the Author):

This paper is on an important subject, aiming to unravel the molecular mechanisms of how wild-derived mice, such as CIM, are resistant to infection with typically highly mouse virulent *T. gondii* strains. The authors show specific interactions between a CIM mice-specific *Irgb2-b1* isoform and ROP5B. This interaction is specific to only the ROP5BRH isoform and not to ROP5ARH and CRH. Interestingly, the N-terminal binding site of *Irgb2-b1*CIM is highly polymorphic and has undergone positive selection in comparison to the C-terminus. Additionally, the authors find that ROP5B interactions with *Irgb2-b1* is specific to the RH strain of *T. gondii* because VAND ROP5 isoforms did not bind to *Irgb2-b1*. Moreover, in strain VAND there is significant interaction between ROP5 and *Irga6-P**.

The proposed model outlined in figure 7 is interesting and potentially compelling, and if proven true this would indeed be a very exciting result. The idea that specific Rop5 paralogs might interact with specific mouse IRG proteins has been proposed but not yet proven. This paper is a step in that direction but there are some important missing pieces for the model.

1. Little information on the sequences unique to ROP5B[rh] that drive the interaction with *Irgb2-b1*CIM.
2. Little information on molecular basis for VAND ROP5 paralogs not binding to *Irgb2-b1*CIM, nor any data on VAND ROP5 interactions with ROP18 as shown in the model (ROP18 is discussed only near the end)
3. Lack of structural or biochemical data (via structural analysis, homology modeling, or mutagenesis) to identify which residues in *rop5* and/or *irgb2-b1* drive these interesting strain- and species-specific differences.

The argument that *Irgb2-b1* interactions are key to differences in resistance in CIM mice is compelling, although some of the conclusions could be bolstered by further experiments to fully establish ROP5 paralog-specific interactions with mouse IRGs. Since ROP5 is a multicopy locus, interactions should be confirmed when only one ROP5 paralog is expressed in RH-ROP5 null parasites. If this is not possible, I would recommend another route by using ROP5 paralog-specific antibodies.

Another critique is the lack of any IFA data in the main figures and sometimes this was also lacking in the supplements. While they may seem redundant to the authors, showing some representative images would help bolster the argument.

Terminology at times is not quantitative when it should be. Also in cases where there are no statistical analyses (e.g., the PCA assay) the authors should explain how they decide if a given assay result was positive or negative.

There is not adequate data about how rop5 paralog sequences were obtained and how they were validated. The sequences used should be shown in the manuscript and aligned. An alignment of the IRG proteins and better illustrations of which parts of the sequences were used for the chimera studies is also important.

Specific comments:

Introduction:

Line 62: I would recommend not describing the genomes of Types 1,2 and 3 "homogenous". There are striking genotypic differences between the genomes even when discounting polymorphic loci like ROP5.

Line 78: change "Irgm1-Irgm3" to "Irgm1, 2, and 3".

Results:

Line 152: citation for "we have already provided evidence" (in addition to the citation provided a few sentences down). also the wording could be improved perhaps "Previously published work..." "our recent work"...etc.

line 165: "almost identical" is not quantitative. wording should be changed. same with "scarcely" (line 168)

line 168: the data in fig. 1C is not a phosphorylation assay but rather an assay to show whether p-ated Irga6 binds to the vacuole. text should be altered to reflect that.

line 170: “approached the same level” is again not quantitative and vague. description of results should focus on clear statistics-based findings instead of terms such as these.

Line 189: “virulent” should be changed to “mouse virulent” in reference to RH since there are dramatic host differences in virulence of this and other *T. gondii* strains (e.g., RH is not lethal to rats)

Line 232: for data from the PCA assay it is unclear how differences are being evaluated. again, non-quantitative terms and statements are made and this needs to be rectified (terms like “almost completely abrogated”, etc.)

Figure 4d: unsure what “1” and “3” labels are at top of graph. also a and b labels meant to indicate replicates should be moved or changed since they are easily confused with statistical significance results.

Labeling of the Y2H plates in terms of where the controls are could be improved by making the lines more clear...also in many cases the labels do not clarify which allele is being tested. e.g., it can be difficult to interpret supp figure 5. Is it always VAND GRA7? should be able to see it on the figure.

The authors successfully made two *Irgb2-b1* ko CIM cell lines and proved that knocking out this particular IRG protein left the ko cells incapable of restricting RH *T. gondii* growth. Additionally, they were able to complement *Irgb2-b1*CIM in the T17 line and restored the resistance phenotype.

Figure 1B: Please include representative IFA images in at least some primary figures as well as in the supplements.

The authors identified an interaction between *Irgb2-b1*CIM and ROP5RH isoform B through both immunoprecipitation experiments and Y2H analysis. They found that recruitment of *Irgb2-b1* to the PVM is dependent on ROP5 expression.

Figure 2A and B: There is no negative control blot for this figure (probing for proteins that shouldn't come down in the IP).. The authors show that *Irgb2-b1*CIM and ROP5 pull down with one another, but they should include an additional western blot showing that an additional protein not predicted

to bind to either of these does not pull down. A final note that I might have missed, but I do not know what the star is indicating in 2A. Please define.

Line 203-210: The three ROP5 isoforms are referenced multiple times throughout the paper. I would suggest citing the sequences included in Genbank to highlight these three polymorphic paralogs. Additionally, please state that these are the RH isoforms of ROP5. as there is extensive interstrain polymorphism at this locus as well.

If expressing individual ROP5 paralogs was possible for the Y2H, it would seem possible to blot for ROP5B using a ROP5B-specific antibody in immunoprecipitation assays. While Y2H analysis may be indicative of what is happening at the host-parasite interface, ROP5B-specific antibody will solidify this conclusion.

An additional experiment that I would recommend is expressing individual ROP5A, B, and C paralogs in RH Δ rop5 parasites and conducting pull downs similar to Figure 2A and B. Irgb2-b1CIM should only pull down with ROP5B and not the others. This will validate specificity in vivo in addition to in vitro Y2H analysis. An important experiment to bolster the conclusions of the paper.

The authors were able to show that a chimeric Irgb2-b1 construct with the N-terminus of Irgb2-b1CIM with the α D/H4 and C-terminus from Irgb2-b1BL/6 binds only to ROP5C indicating the requirement of the Irgb2-b1CIM α D/H4 for ROP5B interaction.

Lines 220-228: These lines in parallel with Figure 3B are a little confusing. It appears that the “evolutionary hot spot” region, α D/H4, is required for Irgb2-b1CIM binding to ROP5. This region is described as being at the N-terminus of the protein as that is the region that actually pulled down ROP5 when fused to GST (Figure 2A). However, this is not clear in the domain diagram in the figure. I would recommend adding residue numbers to this figure to better describe the exact borders of the chimeric construct.

Figure 3E: I would recommend specifying what WT means here. I am assuming it is the T17 + Irgb2-b1CIM from what I can deduce from Figure 4C, but it should be noted.

The authors investigated the specificity of ROP5C (and not ROP5B and A) binding to Irgb2-b1BL/6. They were able to show that ROP5C binds specifically to Irgb2-b1BL/6 through Y2H analysis and that expressing only Irgb2-b1BL/6 does not restore CIM growth inhibition phenotype. Additionally, they found that expressing Irgb2-b1 chimera in T17 ko cells showed significantly less loading of IRGs at the PVM and matched the phenotype of T17 + Irgb2-b1BL/6, suggesting ROP5C does not contribute significantly to virulence in CIM mice.

I recommend a different title for this section. The title as it stands seems quite misleading because you did not actually test “virulence” of specifically the ROP5B isoform in CIM mice. This comment goes back to a previous recommendation of expressing individual ROP5 isoforms in RH Δ ROP5 parasites to really draw conclusions about in vivo specific binding and virulence in mice.

The authors investigated two South American *T. gondii* strains (AS28 and VAND) for their in vitro growth inhibition and in vivo virulence. Both strains were lethal in BL/6 and CIM mice and only VAND showed no growth inhibition after IFN γ stimulation while AS28 had an intermediate phenotype compared to VAND and RH. They found that VAND showed an increase in Irga6 phosphorylation which previously correlated with low Irgb2-b1 loading at the PVM. Importantly, the authors found that VAND did not pull down or interact with any of the ROP5 isoforms through both colP and Y2H analyses.

Figure 6C: See above notes for comments related to pull down controls.

Since the authors have at least some forms of VAND rop5 sequences can they please examine the sequences to identify putative polymorphisms that might be relevant? Or at least discuss how many there are that could be responsible for the lack of IRG binding?

Discussion:

Lines 354-357: As ROP5 is an expanded and diversified locus among Types 1, 2 and 3, please indicate that these are RH ROP5 isoforms. You mention it in the discussion and materials, but please also include it in the results.

Lines 375-377: I would recommend the authors including an alignment of this polymorphic region for the reader to see the evolution of the N-terminus in comparison to the remainder of the Irgb2-b1 protein. This should include a few of the Irgb2-b1 isoforms in addition to Irgb2-b1CIM and Irgb2-b1BL/6. This would allow the reader to be better convinced of this evolutionary hot spot that seems to be required for binding ROP5 in a paralog-specific manner.

Line 423: not sure if we can say that the IRG system has been lost in humans compared to mice? (as opposed to gained in mice?)

lines 430-432: the discussion of 'old' vs. 'new world' for the mice and toxoplasma seems to be a bit thin, is not cited and overly speculative.

Methods: how were the VAND isoforms obtained? through PCR? no methods are provided but this is crucial for these particular gene classes which are subject to chimerism during amplification. concern is that the isoforms being evaluated have no real representative in nature. also these sequences if new should be deposited in genbank. if not new accession numbers should be provided.

Reviewer #2 (Remarks to the Author):

In the present study, Steinfeldt and colleagues analyzed the molecular mechanisms of how cells from wild-derived *Mus musculus castaneus* CIM mice, which are resistant to virulent RH *Toxoplasma* strain, show defense against the *Toxoplasma* strain. In detail, CIM cells express a tandem IRG called *Irgb2-b1*, which confers resistance to the RH *Toxoplasma* strain. In addition, the authors demonstrated the interaction of *Irgb2-b1* and ROP5, which is a major *Toxoplasma* virulence effector secreted from rhoptry. Furthermore, they analyzed and identified a specific region called α d and H4 spots in *Irgb2-b1* as a determinant for resistance found in CIM (resistant) and C57BL/6 (susceptible). Although CIM mice were susceptible to South America-derived *Toxoplasma* strains VAND and AS28, ROP5 from VAND failed to interact with *Irgb2-b1*, accounting for the more proceeding evolution of ROP5 than *Irgb2-b1*. Overall, the study extends their previous unique work focusing on host IRG and *Toxoplasma* virulence factors, and shows a new mechanism on *Irgb2-b1* targeting on ROP5 as the bottle neck between two organisms. Although this study is potentially of interest, additional complete sets of experiments and description and statistical analysis to support their hypothesis on co-evolution of the IRG and ROP system should be required.

Major comments

1) The statistical analyses of this study are a major concern to me. Throughout the manuscript, the authors state to show representative (or both) data from 2 independent experiments. Many of the experiments were performed with 2 biological replicates. Although it is feasible to show representative data, it is not appropriate to perform significance analysis with the data from technical replicates out of one (or two) experiment (e.g. Fig. 1C, 1D, 1E, 3E, 4C, 4D, 6A, 6B). All data with statistics should be the mean of three biologically independent experiments. This is also clearly stated in the notes of the Nature Communications about statistics reporting. Also, it is not appropriate to use means or error bars with a sample size of 2. The authors should base their significance analysis on the data from at least 3 independent experiments and provide evidence that

the independent experiments show reproducible biological effects. One wonders how significant the results in this paper are if correct statistical methods are applied throughout.

Also, several graphs show no information about statistical significance: Fig. 2D, 3C, 4A, 5A, 5B, 5C. How many times did the author replicate the in vivo experiments in Fig. 5A and 5B? The sample size (number of mice) was too small.

2) Fig.2: *Irgb2-b1* associates with ROP5B and ROP5C but not with ROP5A. What is the difference between ROP5BC and ROP5A? The authors should reveal the molecular mechanism more.

3) Fig.3BC: α D and H4 spots in *Irgb2-b1* decide the difference between the CIM and the C57BL/6 forms. What features of the region determine the difference? Is the region structurally important? How different are they? More experimental data are required.

4) Fig.3BC.: ROP5C associates with *Irgb2-b1* through the domain distinct from ROP5B. Based on the data of CIM-BL6 chimera, the N-terminus may determine the differential binding capacity to ROP5C. What is the detailed difference? More experimental or in silico structural data are required.

5) Fig. 5C: Among South American *Toxoplasma* strains, VAND showed resistance to high and low concentrations of IFN- γ stimulation. In contrast, AS28 showed susceptibility to the high concentration of IFN- γ stimulation. What is the difference between AS28 and VAND? Does the *Irgb2-b1* expression or polymorphisms account for the difference?

6) Fig. 6B: In VAND infected CIM cells, *Irgb2-b1* was not accumulated on the *Toxoplasma* PVs. Where was the *Irgb2-b1* localized? Basically, molecular characterization of *Irgb2-b1* is insufficient. Where is *Irgb2-b1* localized in unstimulated cells? Similar to *Irga6*, is *Irgb2-b1* localized on ER? Or other location? How is the timing of *Toxoplasma* PV loading? Faster than *Irga6*, *Irgb6* and *Irgb10*? Does *Irgb2-b1* form oligomers with other IRGs? Such basic information on *Irgb2-b1* is helpful for readers and should be included in supplementary Figure.

7) Fig. 6C: The molecular size of GRA7 in VAND seems larger than that of RH or AS28. Does this mean the GRA7 modification? Or polymorphisms in VAND GRA7, potentially accounting for the more virulent phenotype than RH or AS28? In general, are ROP5 and GRA7 in VAND recruited to *Toxoplasma* PVs?

8) Fig. 7: For even researchers in the *Toxoplasma* field, it is very uncommon that the RH strain that is very famous for the hyper virulence in laboratory mice such as C57BL/6 is “avirulent” in wild-derived CIM mice. Therefore, the current picture can mislead general Nature Communications readers that the RH strain is simply “avirulent” in all mouse strains. This reviewer highly recommends the authors to draw not only the illustrations in CIM mice but also those in laboratory mice.

9) In terms of co-evolution (or co-adaptation), why do the South American *Toxoplasma* strains such as VAND and AS28 acquire the resistance to *Irgb2-b1*? Are CIM mice derived from South America, since they co-habit there? Regarding the anti-IRG resistance mechanism in *Toxoplasma*, if the authors’ hypothesis is correct, VAND and AS28 have evolved more than RH and ME49. Regarding the anti-*Toxoplasma* ROP5 virulence mechanism in mice, CIM mice have more evolved IRG system (*Irgb2-b1*) than BL6 mice. If the authors should perform molecular phylogenetic analysis using ROP5 sequences of some *Toxoplasma* strains and *Irgb2-b1* sequences of some mice strains, can the author capture the trace of co-evolution (co-adaptation) in the sequences of *Irgb2-b1* and ROP5 in silico?

10) Fig. 2A, 2B, 3A, 5C: The authors should show not only western blot data for immunoprecipitants but also that of inputs (or whole cell lysates). For example, this reviewer cannot decide whether the *Irgb2-b1* interaction with ROP5 works or not because it is possible that ROP5-KO RH infected cells did not express *Irgb2-b1* or that the authors simply failed to immunoprecipitate ROP5 in IFNg(-) lanes in control RH infected cells. Should not omit any control lanes for immunoprecipitation experiments throughout the manuscript. Not only Fig. 2A but also Fig. 2B, 3A and 5C should be perfectly performed without such omission.

Minor points

11) Line 104: The color of “observation” is blue? Should be black.

12) Suppl. Fig. 5: Images of blots were too chopped. The images should show in uncropped.

Reviewer #3 (Remarks to the Author):

Steinfeldt and colleagues present an interesting new detailed investigation into the molecular interplay between a host resistance protein, an IRG - *Irgb2-b1* – and isoforms of the *Toxoplasma gondii* virulence protein ROP5. They study the difference of the interaction between these two factors in cells derived from inbred laboratory mice versus wild mice in correlation to ROP5 isoforms.

Their results afford the speculative conclusion that certain *T. gondii* strains have co-evolved with wild mice host resistance factors.

While I find the study interesting and convincing, the presentation of the data is incomplete and lacks proper explanation in foremost the text and figure legends (see comments below). Most concerning is the number of biological repeats and the statistical significance derived from them (also see below). With regards to the manuscript text, it will be very hard to follow for a non-*T. gondii* expert. I suggest being mindful of the nomenclature of the IRGs and calling it for example by subscript of the corresponding murine strain throughout. Readers will otherwise get lost easily!

General comments:

The manuscript needs to be proof-read and curated by a native English speaker.

The way technical and biological repeats are conducted and presented is unusual. All data has to be collected in biological triplicate (often only duplicate) and if possible those in technical triplicate (duplicate is often the case). To present statistical errors, triplicate data are required.

Instead of presenting all biological repeats, it is less confusing and more standard to present the merged biological triplicate with error bars. It is not apparent to me how error bars are derived from duplicate samples (e.g. Fig 1D and E, Fig 4D, Fig 5C etc). All statistical methods to derive errors and significance should be defined in the Methods section and mentioned in the relevant Figure legend. It is for example not easy to understand how the statistical differences denoted in Fig 1 are defined – all compared to wt? Each biological replicate compared separately? How is that possible?

For the Figures containing yeast two hybrid data, BlaCm BlaN and BD, AD need to be defined in Figure Legend and not just in the Materials and Methods.

Don't call anything a pull-down, that is lab slang.

Fig 1:

I would change the wording of the interpretation of Fig 1C. The two *Irgb2-b1* ko CIM lines “approach” the same levels of pT108-phosphorylated *Irga6* loading onto the PV as B6 cell lines the authors claim. It will be interesting to see what level of rescue can be achieved adding another

biological repeat. At the moment, eye balling the results, it looks like there is a ca 75% rescue in the KO compared to a B6 wt. And that difference would still be statistically significant.

It is important to present the raw FACS data of the replication assay used in Fig 1 as Supplementary Data. It is also crucial to mention in the Materials what MOI was used. After 24h, RH has replicated a number of times, so theoretically multiple peaks in a histogram should have been observed. Was this the case?

Why is the CIM mutant cell line data sometimes compared to B6 wt and in other panels not?

Write MOI 1 and MOI 3 on the Figure, not just 1 and 3.

Figure legend:

B: "No Irgb2-b1 positive vacuoles were found in Irgb2-b1 ko cells." yet there are 10% visible in the T17 line.

D and E: Should be U/ml and not just ml⁻¹.

Fig 2:

What is star in blot? Put in legend.

A and B: denote on the Figure which antibody was used for IP and which for IB and spell this out in the legend as well. As it is presented now, it is confusing to follow.

Fig 3:

Why were the chimera complemented cells not used for IP experiments to IP Rop5?

E: Rather than just presenting intensity of loaded PVs, %age of loaded PVs should also be counted (like in Fig 1). Essentially, the authors just changed their read-out here. It is very different to assess the number of PVs coated versus the thickness of the coat. State why these different read-outs are used.

Fig 4:

A: How many biological repeats of this assay were performed? Is this a representative of 3?

D: Again, this data has to be presented as the pooled data from 3 biological replicates in order to generate any statistics. Like this, it is essentially raw data of two repeats presented in a very confusing way.

Fig 5:

A and B: This experiment should be conducted three times and not just once with 5 mice.

C: Are these biological triplicates or technical triplicates?

Figure legend says MOI 1 or 3 – which one is shown in the Figure?

Fig 6:

Why is 200U/ml IFN γ now used? Before the amount of IFN γ was lower.

Figure legend A: "... protein intensities are reduced...." That is compared to what?

Again, do not show two representative experiments, but rather the pooled biological triplicate repeats.

Responses to referees' comments

Our responses are interdigitated with the referee comments, in a different font and color.

Before we respond to the referee's comments point-by-point we would like to correct an error that we accidentally made and that has not been addressed in the otherwise very thoughtful reviews.

In Fig. 6D, we analysed Irgb2-b1_{CIM} binding to VAND ROP5 isoforms A, B1, B2 and B3. For none of these 4 isoforms binding to Irgb2-b1_{CIM} was found. This is in agreement with results shown in Fig. 6A, B, C and Suppl. Fig.11. However, as controls in Fig. 6D, we showed VAND GRA7 binding to VAND ROP5A, B1, B2 and B3. Also in this case, no binding is visible. This is a clear mistake and we put this panel accidentally. Binding of VAND GRA7 to VAND ROP5A, B1, B2 and B3 could repeatedly be detected in the YTH screens. This is already shown in the old Suppl. Fig. 6 (now Suppl. Fig. 10) for VAND ROP5B3.

We corrected Fig. 6D and now show one representative example for VAND GRA7 binding to all VAND ROP5 isoforms.

We deeply apologize for this mistake.

Reviewer #1 (Remarks to the Author):

This paper is on an important subject, aiming to unravel the molecular mechanisms of how wild-derived mice, such as CIM, are resistant to infection with typically highly mouse virulent *T. gondii* strains. The authors show specific interactions between a CIM mice-specific Irgb2-b1 isoform and ROP5B. This interaction is specific to only the ROP5BRH isoform and not to ROP5ARH and CRH. Interestingly, the N-terminal binding site of Irgb2-b1_{CIM} is highly polymorphic and has undergone positive selection in comparison to the C-terminus. Additionally, the authors find that ROP5B interactions with Irgb2-b1 is specific to the RH strain of *T. gondii* because VAND ROP5 isoforms did not bind to Irgb2-b1. Moreover, in strain VAND there is significant interaction between ROP5 and Irga6-P*.

The proposed model outlined in figure 7 is interesting and potentially compelling, and if proven true this would indeed be a very exciting result. The idea that specific Rop5 paralogs might interact with specific mouse IRG proteins has been proposed but not yet proven. This paper is a step in that direction but there are some important missing pieces for the model.

1. Little information on the sequences unique to ROP5B[rh] that drive the interaction with Irgb2-b1_{CIM}.

We are thankful that this reviewer draw our attention to this. We now provide a phylogenetic analysis of *rop5* sequences and an alignment of amino acids within the region that was previously shown to mediate the binding of ROP5 to Irga6 and is the predicted interface for Irgb2-b1 binding (Fig. 6). Both analyses have been done before and the respective references are given in the text.

2. Little information on molecular basis for VAND ROP5 paralogs not binding to Irgb-b1_{CIM}, nor any data on VAND ROP5 interactions with ROP18 as shown in the model (ROP18 is discussed only near the end)

The answer to this is partially provided above (1). In addition, we replaced the respective sequence in VAND ROP5B3 with the RH ROP5B interface necessary for Irga6 binding (ROPChimera) and investigated binding to Irgb2-b1_{CIM}. The PCA shows binding of Irgb2-b1_{CIM} to RH ROP5B and ROP5Chimera but not to VAND ROP5B3 (Suppl. Fig. 11).

The reviewer is correct, we don't have experimental data for the interaction of VAND ROP5 and VAND ROP18 that is depicted in our model (Fig. 7). We clearly state now in the respective figure legend that this interaction has to be confirmed by appropriate experimental approaches (Fig. 7).

Another interaction drawn in the model is VAND GRA7 binding to VAND ROP5. This interaction could be demonstrated by YTH analysis for all VAND ROP5 isoforms (Fig. 6D). However, we would be glad to defer this analysis, since the structural basis for the interaction between ROP18 and ROP5 or GRA7 and ROP5 is unknown even for the "classical" strains.

3. Lack of structural or biochemical data (via structural analysis, homology modeling, or mutagenesis) to identify which residues in *rop5* and/or *irgb2-b1* drive these interesting strain- and species-specific differences.

An alignment of amino acids within the region that was previously shown to mediate the binding of RH ROP5 to Irga6 and is the predicted interface for Irgb2-b1 binding is now shown in Fig.6E including all VAND ROP5 isoforms. We also show phylogenetic analysis of *Irgb2* sequences and an alignment of amino acids within the polymorphic region that was previously shown to mediate the binding of Irga6 to ROP5 (Fig. 4E). This analysis is not yet complete, since multiple residues vary in the contact regions, but the alignment establishes where the focus will lie, which we hope will be a sufficient level of detail for the present study.

The argument that *Irgb2-b1* interactions are key to differences in resistance in CIM mice is compelling, although some of the conclusions could be bolstered by further experiments to fully establish ROP5 paralog-specific interactions with mouse *Irgs*. Since ROP5 is a multicopy locus, interactions should be confirmed when only one ROP5 paralog is expressed in RH-ROP5 null parasites. If this is not possible, I would recommend another route by using ROP5 paralog-specific antibodies.

The reviewer raised an interesting point and we have already thought about that before. The antibody used in this study to detect ROP5 is isoform specific but as we show in Suppl. Fig. 6, it can detect ROP5A and ROP5B but not ROP5C. Therefore, it can not be used to discriminate between the three ROP5 paralogs.

Likewise, strains that do express single ROP5 paralogs in a *rop5* ko background are not available to us and we don't have enough experience with the generation of *T. gondii* ko and/or transgenic strains to create the necessary tools and get results within such a short time frame. However, in a previous study (Fleckenstein et al., 2012, PlosBiology), we already used a transgenic *T. gondii* strain that expresses one copy of ROP5A and ROP5B (RH Δ *rop5*+A/B) as HA- or FLAG-tagged fusion proteins (HA-ROP5A/FLAG-ROP5B). Here we have now used both strains to demonstrate binding of ROP5B to *Irgb2-b1*_{CIM} but not *Irgb2-b1*_{BL/6} and *Irgb2-b1*_{Chimera} in infected cells after IP with anti-*Irgb2-b1* (Suppl. Fig. 11).

Another critique is the lack of any IFA data in the main figures and sometimes this was also lacking in the supplements. While they may seem redundant to the authors, showing some representative images would help bolster the argument. Representative fluorescent images for Fig. 1B, 1C and Fig. 6B are now shown in Suppl. Fig. 1, Suppl. Fig. 2 and Suppl. Fig.8 respectively.

Terminology at times is not quantitative when it should be. Also in cases where there are no statistical analyses (e.g., the PCA assay) the authors should explain how they decide is a given assay result was positive or negative.

All underlying statistical methods are now stated in the respective figure legends and the Methods section of the manuscript.

There is not adequate data about how *rop5* paralog sequences were obtained and how they were validated. The sequences used should be shown in the manuscript and aligned. an alignment of the IRG proteins and better illustrations of which parts of the sequences were used for the chimera studies is also important.

All *rop5* sequences were obtained from previous publications. These references are now given in "*Plasmid Constructs*" in the Methods section of the manuscript.

Specific comments:

Introduction:

Line 62: I would recommend not describing the genomes of Types 1,2 and 3 "homogenous". There are striking genotypic differences between the genomes even when discounting polymorphic loci like ROP5.

The reviewer is correct. We deleted the term "otherwise very homogenous".

Line 78: change "Irgm1-Irgm3" to "Irgm1, 2, and 3".

Changed now.

Results:

Line 152: citation for "we have already provided evidence" (in addition to the citation provided a few sentences down). also the wording could be improved perhaps "Previously published work..." "our recent work"...etc.

Reference added now and wording changed to "Our own recent work demonstrates".

line 165: "almost identical" is not quantitative. wording should be changed. same with "scarcely" (line 168)

Wording changed to "identical" and "phosphorylation is almost completely inhibited".

line 168: the data in fig. 1C is not a phosphorylation assay but rather an assay to show whether p-ated Irga6 binds to the vacuole. text should be altered to reflect that.

Phosphorylation of Irga6 in virulent strain infections happens at the PVM (Fentress et al., 2012, Cellular Microbiology). We use an antibody that specifically detects p(T108)Irga6 at the PVM and therefore we think that this assay can actually be considered as a phosphorylation assay.

However, the second part of this specific sentence does in fact clearly state that we are looking for vacuoles carrying p(T108)Irga6.

If the reviewer agrees, we would like to keep our text as it is.

line 170: "approached the same level" is again not quantitative and vague. description of results should focus on clear statistics-based findings instead of

terms such as these.

The reviewer is certainly right and we are sorry that we have overlooked this specification in the guidelines of Nature communications. We now show the mean of at least three biologically independent experiments.

Wording changed to "...are comparable to frequencies in BL/6 cells".

Line 189: "virulent" should be changed to "mouse virulent" in reference to RH since there are dramatic host differences in virulence of this and other *T. gondii* strains (e.g., RH is not lethal to rats)

Changed.

Line 232: for data from the PCA assay it is unclear how differences are being evaluated. again, non-quantitative terms and statements are made and this needs to be rectified (terms like "almost completely abrogated", etc.)

We now show biological triplicates for all PCA assays: ROP5A/B/C vs *Irgb2-b1*_{CIM} (Fig. 2), ROP5A/B/C vs *Irgb2-b1*_{Chimera} (Fig. 3) and ROP5A/B/C vs *Irgb2-b1*_{BL/6} (Fig. 4). Statistical significances are indicated in the figures and the method is now stated in the respective figure legends and the Methods section of the manuscript.

Figure 4d: unsure what "1" and "3" labels are at top of graph. also a and b labels meant to indicate replicates should be moved or changed since they are easily confused with statistical significance results.

"1" and "3" is now changed to "MOI 1" and "MOI 3". Labels "a" and "b" indicating two independent experiments are now deleted and the mean of three independent experiments is shown.

Labeling of the Y2H plates in terms of where the controls are could be improved by making the lines more clear...also in many cases the labels do not clarify which allele is being tested. e.g., it can be difficult to interpret supp figure 5. Is it always VAND GRA7? should be able to see it on the figure.

Lines are more accentuated now and labeling is more comprehensible.

The authors successfully made two *Irgb2-b1* ko CIM cell lines and proved that knocking out this particular IRG protein left the ko cells incapable of restricting RH *T. gondii* growth. Additionally, they were able to complement *Irgb2-b1*_{CIM} in the T17 line and restored the resistance phenotype.

Figure 1B: Please include representative IFA images in at least some primary figures as well as in the supplements.

Representative fluorescent images for Fig. 1B, 1C and Fig. 6B are now shown in Suppl. Fig. 1, Suppl. Fig. 2 and Suppl. Fig. 8 respectively.

The authors identified an interaction between Irgb2-b1CIM and ROP5RH isoform B through both immunoprecipitation experiments and Y2H analysis. They found that recruitment of Irgb2-b1 to the PVM is dependent on ROP5 expression.

Figure 2A and B: There is no negative control blot for this figure (probing for proteins that shouldn't come down in the IP).. The authors show that Irgb2-b1CIM and ROP5 pull down with one another, but they should include an additional western blot showing that an additional protein not predicted to bind to either of these does not pull down.

For both, Fig. 2A and Fig. 2B, the appropriate Western blot controls demonstrating equal expression levels of respective proteins in the lysates are shown.

According to our understanding, the RH Δ rop5 is the optimal control for both experiments for the following reasons:

In the Co-IP (Fig. 2A), where anti ROP5 antibody 3E2 is used to immunoprecipitate ROP5, it demonstrates a) no unspecific binding of 3E2 to Irgb2-b1 b) no unspecific binding of 3E2 to another protein interacting with Irgb2-b1.

In the pull-down (Fig. 2B), it demonstrates that the band detected in RH Δ hxgprt is indeed ROP5 and not another protein bound to Irgb2-b1 and unspecifically recognized by the anti ROP5 antibody.

With respect, we think that probing for proteins that shouldn't come down in the IP and/or Western blot is not necessary to demonstrate specificity.

A final note that I might have missed, but I do not know what the star is indicating in 2A.

We apologize for this mistake. A star, indicating unspecific bands, is specified now in the figure legend.

Line 203-210: The three ROP5 isoforms are referenced multiple times throughout the paper. I would suggest citing the sequences included in Genbank to highlight these three polymorphic paralogs. Additionally, please state that these are the RH isoforms of ROP5. as there is extensive interstrain polymorphism at this locus as well.

The reviewer is right. References for all *rop5* sequences are now given in the text and in "*Plasmid Constructs*" in the Methods section of the manuscript.

We changed the text to indicate RH-derived ROP5 isoforms.

If expressing individual ROP5 paralogs was possible for the Y2H, it would seem possible to blot for ROP5B using a ROP5B-specific antibody in immunoprecipitation assays. While Y2H analysis may be indicative of what is happening at the host-parasite interface, ROP5B-specific antibody will solidify this conclusion.

The anti ROP5 antibody (3E2) that was used in the present study detects ROP5A and B but not ROP5C. Therefore, it can not be used to discriminate between the three ROP5 paralogs.

An additional experiment that I would recommend is expressing individual ROP5A, B, and C paralogs in RH Δ rop5 parasites and conducting pull downs similar to Figure 2A and B. Irgb2-b1CIM should only pull down with ROP5B and not the others. This will validate specificity in vivo in addition to in vitro Y2H analysis. An important experiment to bolster the conclusions of the paper.

This point has been already addressed and the answer is the same: We really would like to have implemented this experiment in our paper but unfortunately a *T. gondii* strain that exclusively expresses single ROP5 isoforms A, B or C are not available to us. Instead, we used a transgenic *T. gondii* strain that expresses HA-ROP5A and FLAG-ROP5B to demonstrate binding of ROP5B to Irgb2-b1_{CIM} but not Irgb2-b1_{BL/6} and Irgb2-b1_{Chimera} in infected cells after anti-Irgb2-b1 IP (Suppl. Fig. 7).

The authors were able to show that a chimeric Irgb2-b1 construct with the N-terminus of Irgb2-b1CIM with the α D/H4 and C-terminus from Irgb2-b1BL/6 binds only to ROP5C indicating the requirement of the Irgb2-b1CIM α D/H4 for ROP5B interaction.

Lines 220-228: These lines in parallel with Figure 3B are a little confusing. It appears that the “evolutionary hot spot” region, α D/H4, is required for Irgb2-b1CIM binding to ROP5. This region is described as being at the N-terminus of the protein as that is the region that actually pulled down ROP5 when fused to GST (Figure 2A). However, this is not clear in the domain diagram in the figure. I would recommend adding residue numbers to this figure to better describe the exact borders of the chimeric construct.

We modified Fig. 3B accordingly.

Figure 3E: I would recommend specifying what WT means here. I am assuming it is the T17 + Irgb2-b1CIM from what I can deduce from Figure 4C, but it should be noted.

Text modified accordingly.

The authors investigated the specificity of ROP5C (and not ROP5B and A) binding

to *Irgb2-b1BL/6*. They were able to show that ROP5C binds specifically to *Irgb2-b1BL/6* through Y2H analysis and that expressing only *Irgb2-b1BL/6* does not restore CIM growth inhibition phenotype. Additionally, they found that expressing *Irgb2-b1* chimera in T17 ko cells showed significantly less loading of IRGs at the PVM and matched the phenotype of T17 + *Irgb2-b1BL/6*, suggesting ROP5C does not contribute significantly to virulence in CIM mice.

I recommend a different title for this section. The title as it stands seems quite misleading because you did not actually test “virulence” of specifically the ROP5B isoform in CIM mice. This comment goes back to a previous recommendation of expressing individual ROP5 isoforms in RHΔROP5 parasites to really draw conclusions about in vivo specific binding and virulence in mice.

The reviewer is correct. We changed the title of the section to “ROP5B is the predominant ROP5 isoform responsible for *T. gondii* virulence in CIM and laboratory mice *in vitro*.”

The authors investigated two South American *T. gondii* strains (AS28 and VAND) for their in vitro growth inhibition and in vivo virulence. Both strains were lethal in BL/6 and CIM mice and only VAND showed no growth inhibition after IFN γ stimulation while AS28 had an intermediate phenotype compared to VAND and RH. They found that VAND showed an increase in *Irga6* phosphorylation which previously correlated with low *Irgb2-b1* loading at the PVM. Importantly, the authors found that VAND did not pull down or interact with any of the ROP5 isoforms through both coIP and Y2H analyses.

Figure 6C: See above notes for comments related to pull down controls.

Appropriate Western blot controls demonstrating equal expression levels of respective proteins in the lysates are now shown in Fig.6.

Since the authors have at least some forms of VAND *rop5* sequences can they please examine the sequences to identify putative polymorphisms that might be relevant? Or at least discuss how many there are that could be responsible for the lack of IRG binding?

An alignment of amino acids is now shown in Fig. 6, including all VAND ROP5 isoforms, for the region that was previously shown to mediate the binding of RH ROP5 to *Irga6* and is the predicted interface for *Irgb2-b1* binding.

Discussion:

Lines 354-357: As ROP5 is an expanded and diversified locus among Types 1, 2 and 3, please indicate that these are RH ROP5 isoforms. You mention it in the

discussion and materials, but please also include it in the results.

We modified the text accordingly.

Lines 375-377: I would recommend the authors including an alignment of this polymorphic region for the reader to see the evolution of the N-terminus in comparison to the remainder of the *Irgb2-b1* protein. This should include a few of the *Irgb2-b1* isoforms in addition to *Irgb2-b1CIM* and *Irgb2-b1BL/6*. This would allow the reader to be better convinced of this evolutionary hot spot that seems to be required for binding ROP5 in a paralog-specific manner.

These alignments have been published before and the reference is given in the text. We now show this phylogenetic analysis of *Irgb2* sequences and an alignment of amino acids within the polymorphic region that was previously shown to mediate the binding of *Irga6* to ROP5 (Fig. 4E).

Line 423: not sure if we can say that the IRG system has been lost in humans compared to mice? (as opposed to gained in mice?)

The answer to this is stated in detail in the following two publications:

Bekpen et al., 2005, *Genome Biol.*

Bekpen et al., 2009, *Plos Genet.*

lines 430-432: the discussion of 'old' vs. 'new world' for the mice and toxoplasma seems to be a bit thin, is not cited and overly speculative.

The last part of the discussion attempts to express our view that the ROP5 kinase complex alleles of South American *T. gondii* strains have arisen as a result of contact with SA mammals, resulting in a mutually tolerable balance "you don't kill me, I won't kill you", but not with *M. musculus*, where things don't seem to be in balance yet, at least for the mouse and *T. gondii* strains asayed. This is perhaps a speculation, but it is important to allow speculation in the discussions of papers.

Methods: how were the VAND isoforms obtained? through PCR? no methods are provided but this is crucial for these particular gene classes which are subject to chimerism during amplification. concern is that the isoforms being evaluated have no real representative in nature. also these sequences if new should be deposited in genbank. if not new accession numbers should be provided.

All *rop5* sequences were obtained from previous publications. These references are now given in "Plasmid Constructs" in the Methods section of the manuscript. Sequences for all VAND *rop5* isoforms were provided for IDT (Integrated DNA Technologies) and after synthesis subcloned from the IDT vectors using PCR and validated by sequencing. This is stated in

“Plasmid Constructs” in the Methods section.

Reviewer #2 (Remarks to the Author):

In the present study, Steinfeldt and colleagues analyzed the molecular mechanisms of how cells from wild-derived *Mus musculus castaneus* CIM mice, which are resistant to virulent RH *Toxoplasma* strain, show defense against the *Toxoplasma* strain. In detail, CIM cells express a tandem IRG called *Irgb2-b1*, which confers resistance to the RH *Toxoplasma* strain. In addition, the authors demonstrated the interaction of *Irgb2-b1* and ROP5, which is a major *Toxoplasma* virulence effector secreted from rhoptry. Furthermore, they analyzed and identified a specific region called α d and H4 spots in *Irgb2-b1* as a determinant for resistance found in CIM (resistant) and C57BL/6 (susceptible). Although CIM mice were susceptible to South America-derived *Toxoplasma* strains VAND and AS28, ROP5 from VAND failed to interact with *Irgb2-b1*, accounting for the more proceeding evolution of ROP5 than *Irgb2-b1*. Overall, the study extends their previous unique work focusing on host IRG and *Toxoplasma* virulence factors, and shows a new mechanism on *Irgb2-b1* targeting on ROP5 as the bottle neck between two organisms. Although this study is potentially of interest, additional complete sets of experiments and description and statistical analysis to support their hypothesis on co-evolution of the IRG and ROP system should be required.

Major comments

1) The statistical analyses of this study are a major concern to me. Throughout the manuscript, the authors state to show representative (or both) data from 2 independent experiments. Many of the experiments were performed with 2 biological replicates. Although it is feasible to show representative data, it is not appropriate to perform significance analysis with the data from technical replicates out of one (or two) experiment (e.g. Fig. 1C phospho *Irga6* frequencies, 1D, 1E replication FACS, 3E *b2-b1* intensities, 4C *b2-b1* intensities, 4D replication FACS, 6A phospho *Irga6* intensities, 6B *b2-b1* intensities). All data with statistics should be the mean of three biologically independent experiments. This is also clearly stated in the notes of the Nature Communications about statistics reporting. Also, it is not appropriate to use means or error bars with a sample size of 2. The authors should base their significance analysis on the data from at least 3

independent experiments and provide evidence that the independent experiments show reproducible biological effects. One wonders how significant the results in this paper are if correct statistical methods are applied throughout. The reviewer is certainly right and we are sorry that we have overlooked this specification in the guidelines of Nature communications. We now show the mean of at least three biologically independent experiments. Significances remain largely unchanged in all the cases.

Also, several graphs show no information about statistical significance: Fig. 2D, 3C, 4A, 5A, 5B, 5C. How many times did the author replicate the *in vivo* experiments in Fig. 5A and 5B? The sample size (number of mice) was too small.

For *in vivo* experiments, VAND infections (Fig. 5A) have been repeated 2 more times and AS28 (Fig. 5B) one more time with higher number of mice respectively. Data are now combined in one diagram.

2) Fig.2: Irgb2-b1 associates with ROP5B and ROP5C but not with ROP5A. What is the difference between ROP5BC and ROP5A? The authors should reveal the molecular mechanism more.

The reviewer is correct. We now provide a phylogenetic analysis of *rop5* sequences and an alignment of amino acids within the region that was previously shown to mediate the binding of ROP5 to Irga6 and is the predicted interface for Irgb2-b1 binding (Fig. 6E). Both analyses have been done before and the respective references are given in the text.

3) Fig.3BC: α D and H4 spots in Irgb2-b1 decide the difference between the CIM and the C57BL/6 forms. What features of the region determine the difference? Is the region structurally important? How different are they? More experimental data are required.

We now again show a phylogenetic analysis of *Irgb2* sequences and an alignment of amino acids within the polymorphic region that was previously shown to mediate the binding of Irga6 to ROP5 (Fig. 4E). Respective references for these analyses are given in the manuscript text.

4) Fig.3BC.: ROP5C associates with Irgb2-b1 through the domain distinct from ROP5B. Based on the data of CIM-BL6 chimera, the N-terminus may determine the differential binding capacity to ROP5C. What is the detailed difference? More experimental or *in silico* structural data are required.

This refers to the point that has been already addressed above. We now

again show a phylogenetic analysis of *Irgb2* sequences and an alignment of amino acids within the polymorphic region that was previously shown to mediate the binding of Irga6 to ROP5 (Fig. 4E).

5) Fig. 5C: Among South American *Toxoplasma* strains, VAND showed resistance to high and low concentrations of IFN- γ stimulation. In contrast, AS28 showed susceptibility to the high concentration of IFN- γ stimulation. What is the difference between AS28 and VAND? Does the *Irgb2*-b1 expression or polymorphisms account for the difference?

For *in vitro* results, this difference in virulence might be explained by residual binding of *Irgb2*-b1 to AS28-derived ROP5 (Fig. 6C). However, these results do not reflect the *in vivo* studies. Here, AS28 may be even more virulent than VAND suggesting the possibility of the existence of additional virulence effects, as already discussed in the paper. Searching for these additional virulence effects can not be part of the present study but demands careful investigation in future.

6) Fig. 6B: In VAND infected CIM cells, *Irgb2*-b1 was not accumulated on the *Toxoplasma* PVs. Where was the *Irgb2*-b1 localized? Basically, molecular characterization of *Irgb2*-b1 is insufficient. Where is *Irgb2*-b1 localized in unstimulated cells? Similar to Irga6, is *Irgb2*-b1 localized on ER? Or other location? How is the timing of *Toxoplasma* PV loading? Faster than Irga6, Irgb6 and Irgb10? Does *Irgb2*-b1 form oligomers with other IRGs? Such basic information on *Irgb2*-b1 is helpful for readers and should be included in supplementary Figure.

The reviewer raised an interesting point. However, the determination of *Irgb2*-b1 localisation in uninfected and infected cells - beyond accumulation at the PVM - requires careful analysis using different markers for various intracellular organelles. These analyses are substantial and are beyond the scope of our paper. It would also not change the conclusion of our study. We can clearly demonstrate that resistance of CIM mice against virulent *T. gondii* strains is mediated by *Irgb2*-b1 targeting of ROP5 at the PVM. The timing of IRG protein loading at the PVM and a hierarchy of that pattern was done for Irga6, Irgb6, Irgb10 and Irgd. Again, these analyses require live cell imaging microscopy and are basically much too extensive to be part of our present study. In recent experiments we have observed somewhat lower numbers of vacuoles loaded with effector IRG proteins Irga6, Irgb6, Irgb10 and Irgd in T17 *Irgb2*-b1 ko cells than in wt cells, consistent with, but far from demonstrating, that effector proteins may interact with *Irgb2*-b1 and thereby gain further affinity for the PVM. This needs further work and is not critical for this study.

7) Fig. 6C: The molecular size of GRA7 in VAND seems larger than that of RH or AS28. Does this mean the GRA7 modification? Or polymorphisms in VAND GRA7, potentially accounting for the more virulent phenotype than RH or AS28?

We agree, the size shift in case of VAND GRA7 is striking and could be explained by either co- or posttranslational modification(s) or more likely due to amino acid polymorphisms already described. Note that even a single amino acid substitution can cause differential running behavior in SDS-PAGE, so the difference does not mean that the slower-running VAND protein is bigger. However, we doubt whether polymorphic differences in GRA7 contribute to virulence differences between *T. gondii* strains because in case of BL/6 Irga6 phosphorylation, mediated by RH ROP5/ROP18/GRA7, GRA7 is only partially contributing to achieve the full virulence potential and both, GRA7 and ROP18, are specific for Irga6 (Hermanns et al., 2016, Cellular Microbiology). On the other hand, the impact of ROP5 on the IRG system was demonstrated to be much more important. Therefore, we think and conclusively demonstrated in the present study that virulence of VAND (and partially AS28) is due to escape from Irgb2-b1 targeting.

In general, are ROP5 and GRA7 in VAND recruited to Toxoplasma PVs?

The whole process of *T. gondii* ROP5/ROP18/GRA7-dependent Irga6 phosphorylation and inactivation was demonstrated to take place at the PVM. Here we show Irga6 phosphorylation in VAND infections at the same threonine residue (T108) using a p(T108)Irga6-specific antibody (558) (Fig. 6A). Therefore, we can expect VAND ROP5, ROP18 and GRA7 to be recruited to the PVM as well. Furthermore, we now show representative images for Irga6 phosphorylation in RH-, VAND- and AS28-infected CIM cells. In this context, a clear localization of AS28 and VAND GRA7 at the PVM is apparent (Suppl. Fig.8).

8) Fig. 7: For even researchers in the Toxoplasma field, it is very uncommon that the RH strain that is very famous for the hyper virulence in laboratory mice such as C57BL/6 is “avirulent” in wild-derived CIM mice. Therefore, the current picture can mislead general Nature Communications readers that the RH strain is simply “avirulent” in all mouse strains. This reviewer highly recommends the authors to draw not only the illustrations in CIM mice but also those in laboratory mice.

We modified the figure accordingly.

9) In terms of co-evolution (or co-adaptation), why do the South American *Toxoplasma* strains such as VAND and AS28 acquire the resistance to *Irgb2-b1*? Are CIM mice derived from South America, since they co-habit there? Regarding the anti-IRG resistance mechanism in *Toxoplasma*, if the authors' hypothesis is correct, VAND and AS28 have evolved more than RH and ME49. Regarding the anti-*Toxoplasma* ROP5 virulence mechanism in mice, CIM mice have more evolved IRG system (*Irgb2-b1*) than BL6 mice. If the authors should perform molecular phylogenetic analysis using ROP5 sequences of some *Toxoplasma* strains and *Irgb2-b1* sequences of some mice strains, can the author capture the trace of co-evolution (co-adaptation) in the sequences of *Irgb2-b1* and ROP5 in silico?

We are happy to review these complex issues.

1. The referee asks why SA strains acquire resistance to *Irgb2-b1*. We do not believe, and do not say, that this has happened. SA *T. gondii* strains evolved their resistance kinases to accommodate the indigenous SA fauna, which does not include *Mus musculus* (Mm), who arrived in SA only 500 years ago. Mm is probably now an evolutionarily significant intermediate host in urban areas of SA in the sense that it participates relatively frequently in transmission between definitive hosts, but if so, only became so with its arrival in the 16th century from the European colonial ships with the domestic cat as the major definitive host. As an evolutionarily significant host, Mm is undoubtedly only one among other, indigenous, species in the SA environment. There is no relevant information about what proportion of *T. gondii* transmission passes through an Mm intermediate host in any ecosystem, interesting though it would be (and hard to come by).

2. The referee says the SA strains have "resistance" to *Irgb2-b1*. We would not support this form of words. All mice probably express a protein from the *Irgb2-b1* locus, though it is highly polymorphic and mice from different parts of the world express different alleles with different expression levels. The *Irgb2-b1* allelic product of CIM is extremely effective in allowing survival against the virulence of the Type I Eurasian strains. The key point, however, is that it is precisely this resistance that makes the CIM mice (and other wild mice carrying this allele) into suitable intermediate hosts for *Toxoplasma*. CIM mice do not die when they are infected by Type I strains, they control the infection and are permissive for the encystment of type I strains in the brain, and thus enable their transmission to another definitive host. Thus the *Irgb2-b1* allele, common across SE Asia, seems to be adaptively coevolved with type I strains, also abundant in that region. To our knowledge a mouse IRG allele that confers sterile immunity on a mouse, ie that eliminates the parasite before encystment, has not yet been

described. Some strains of rat eliminate the parasite completely, but this resistance is not due to the IRG system.

3. VAND and AS28 are SA strains that are highly virulent for Eurasian mouse strains. They are surely coevolved with and become adapted to other indigenous SA intermediate host species in which they are less virulent. They are evidently not coevolved with *Mm* because they kill these hosts, a clearly disadvantageous behaviour for both sides. We cannot say they have “evolved more” as the referee suggests. They are probably in a satisfactory co-evolutionary relationship with indigenous SA species as a result of prolonged cohabitation and coevolution. They simply haven’t had an opportunity or perhaps also a need, to co-evolve with *Mus musculus*, a recent arrival of only 500 years standing and perhaps not (or not yet) even an evolutionarily important host in most of SA.

4. The referee suggests that CIM mice “have a more evolved IRG (b2b1) system than C57BL/6”. We would perhaps rather say that the two strains reflect *Irgb2-b1* alleles that have evolved under different selective pressures. CIM and other SE Asian strains have presumably been under pressure from the high abundance of highly virulent type I Eurasian strains in that geographical area. The relatively poorly expressed (but still intact and capable of interacting with ROP5 allelic products as shown in Fig. 4A, B) *Irgb2-b1* of BL/6 may not be effective against virulent type I strains, both because of its low expression level and its distinct specificity, but type I strains, at least at present, seem to be uncommon in the Western and Northern Eurasian *Toxoplasma* pool. We can perhaps speculate that the sequence of *Irgb2-b1*_{BL/6} was evolved under the pressure of a previously abundant but now rare *T. gondii* strain, and that its current low expression is a response to the relief from that pressure with increasing rarity of the relevant *T. gondii*. Isolation of *T. gondii* from species of the environment typical of the wildcat *F. sylvestrus* in Asia and the Middle East might reveal virulent strains for which well-expressed *Irgb2-b1* (BL/6) might be a suitable resistance allele.

10) Fig. 2A pull down, 2B Co-IP, 3A pull down, 5C: The authors should show not only western blot data for immunoprecipitants but also that of inputs (or whole cell lysates). For example, this reviewer cannot decide whether the *Irgb2-b1* interaction with ROP5 works or not because it is possible that ROP5-KO RH infected cells did not express *Irgb2-b1* or that the authors simply failed to immunoprecipitate ROP5 in IFNg(-) lanes in control RH infected cells. Should not omit any control lanes for immunoprecipitation experiments throughout the

manuscript. Not only Fig. 2A but also Fig. 2B, 3A and 5C should be perfectly performed without such omission.

Appropriate Western blot controls demonstrating equal expression levels of respective proteins in the lysates are now supplemented in Figures 2A, 2B, 3A and 6C.

Minor points

11) Line 104: The color of “observation” is blue? Should be black.

We apologize for this mistake. Corrected now.

12) Suppl. Fig. 5: Images of blots were too chopped. The images should show in uncropped.

Amounts of GST-Irgb2-b1 and GST used for the pull-down experiment are now shown in uncropped images (Suppl. Fig. 10).

Reviewer #3 (Remarks to the Author):

Steinfeldt and colleagues present an interesting new detailed investigation into the molecular interplay between a host resistance protein, an IRG - Irgb2-b1 – and isoforms of the *Toxoplasma gondii* virulence protein ROP5. They study the difference of the interaction between these two factors in cells derived from inbred laboratory mice versus wild mice in correlation to ROP5 isoforms. Their results afford the speculative conclusion that certain *T. gondii* strains have co-evolved with wild mice host resistance factors.

While I find the study interesting and convincing, the presentation of the data is incomplete and lacks proper explanation in foremost the text and figure legends (see comments below). Most concerning is the number of biological repeats and the statistical significance derived from them (also see below). With regards to the manuscript text, it will be very hard to follow for a non-*T. gondii* expert. I suggest being mindful of the nomenclature of the IRGs and calling it for example by subscript of the corresponding murine strain throughout. Readers will otherwise get lost easily!

The reviewer is certainly right. We now show the mean of at least three biologically independent experiments and calculated the statistical significances. The underlying statistical methods are stated in the respective figure legends and Material & Methods.

Concerning the nomenclature of IRG and *T. gondii* proteins, we modified the manuscript text according to the reviewer's suggestion.

General comments:

The manuscript needs to be proof-read and curated by a native English speaker.

The way technical and biological repeats are conducted and presented is unusual. All data has to be collected in biological triplicate (often only duplicate) and if possible those in technical triplicate (duplicate is often the case). To present statistical errors, triplicate data are required.

Instead of presenting all biological repeats, it is less confusing and more standard to present the merged biological triplicate with error bars. It is not apparent to me how error bars are derived from duplicate samples (e.g. Fig 1D and E, Fig 4D, Fig 5C etc). All statistical methods to derive errors and significance should be defined in the Methods section and mentioned in the relevant Figure legend. It is for example not easy to understand how the statistical differences denoted in Fig 1 are defined – all compared to wt? Each biological replicate compared separately? How is that possible?

As already indicated above, we now show the mean of at least three biologically independent experiments and all statistical methods are mentioned in the respective figure legends and defined in Material & Methods.

For the Figures containing yeast two hybrid data, BlaCm BlaN and BD, AD need to be defined in Figure Legend and not just in the Materials and Methods.

In case of PCA, BlaC and BlaN, in case of YTH, pGAD and pGBD, are now defined in the respective figure legends.

Don't call anything a pull-down, that is lab slang.

The "lab slang" referred to by the reviewer is, (perhaps regrettably) already a common place in many eminent journals, including Nature Communications, Journal of Experimental Medicine, EMBO etc..

Here are some prominent recent uses of the phrase:

Protein-Protein Interactions: Pull-Down Assays. Louche A et al. Methods Mol Biol. (2017)

EMBO J. 2017 Feb 1;36(3):361-373. doi: 10.15252/emj.201592426. (in abstract)

Nat Commun. 2016 Oct 3;7:12832. doi: 10.1038/ncomms12832. (in abstract)

Nature. 2011 May 26;473(7348):484-8. doi: 10.1038/nature10016. (in title)

Nature. 2011 May 26;473(7348):461-2. doi: 10.1038/473461a. (in title)

We could consider an alternative designation if the reviewer can suggest one

that is as widely used as “pull-down”, doesn’t simply introduce a new “lab slang” into the literature, and clearly discriminates this procedure from an immunoprecipitation.

Fig 1:

I would change the wording of the interpretation of Fig 1C. The two *Irgb2-b1* ko CIM lines “approach” the same levels of pT108-phosphorylated *Irga6* loading onto the PV as B6 cell lines the authors claim. It will be interesting to see what level of rescue can be achieved adding another biological repeat. At the moment, eye balling the results, it looks like there is a ca 75% rescue in the KO compared to a B6 wt. And that difference would still be statistically significant.

Also in this case, we now show the mean of three biologically independent experiments. Accumulation of p(T108)*Irga6* at the PVM in *Irgb2-b1* ko cell lines is comparable to BL/6 cells (differences are not significant). We changed the manuscript text to “...are comparable to frequencies in BL/6 cells”.

It is important to present the raw FACS data of the replication assay used in Fig 1 as Supplementary Data. It is also crucial to mention in the Materials what MOI was used. After 24h, RH has replicated a number of times, so theoretically multiple peaks in a histogram should have been observed. Was this the case?

A representative image for Fig. 1D is now shown in Suppl. Fig. 3 and the MOI is clearly indicated in the figure and mentioned in the figure legend.

Why is the CIM mutant cell line data sometimes compared to B6 wt and in other panels not?

One representative correlate of *T. gondii* virulence is the phosphorylation of *Irga6*. CIM wt cells are resistant and inhibit *Irga6* phosphorylation because of *Irgb2-b1* binding to ROP5. BL/6 wt cells are susceptible because the allelic *Irgb2-b1* variant can not bind and block ROP5. To clarify this impact of *Irgb2-b1*, *Irgb2-b1* ko cell lines (T17 and i3) have been compared to BL/6 wt cells in Fig. 1C.

In Fig. 4, the correlate of virulence is *T. gondii* replication in different cell lines. *T. gondii* replication is inhibited in CIM wt cells because of *Irgb2-b1* binding to ROP5. This resistance is absent in BL/6 wt cells and in CIM cells in the absence of *Irgb2-b1* (T17) because ROP5 can not be blocked. We want to demonstrate the impact of *Irgb2-b1*_{CIM}, *Irgb2-b1*_{BL/6} and *Irgb2-b1*_{Chimera} in an *Irgb2-b1* ko background. Therefore, these transgenic cell

lines are compared to CIM wt cells.

Write MOI 1 and MOI 3 on the Figure, not just 1 and 3.

Done.

Figure legend:

B: "No *Irgb2-b1* positive vacuoles were found in *Irgb2-b1* ko cells." yet there are 10% visible in the T17 line.

In this experiment we determine the percentage of *Irgb2-b1*-positive vacuoles in CIM wt and *Irgb2-b1* ko cells. In contrast to intensity measurements, this kind of analysis does not consider vacuolar protein amounts which always fluctuate from low to high intensities (e.g. Fig. 3E). Therefore, every evaluation requires definition of the minimal signal intensity to consider a vacuole as positive or negative. All data have been obtained in four independent biological repeats by different experimenters. Every experimenter had a different personal definition for these minimal signal intensity, explaining 5 % (see new Fig. 1B) *Irgb2-b1*-positive vacuoles in T17 cells. But the reviewer is certainly right and we modified the text accordingly.

D and E: Should be U/ml and not just ml⁻¹.

Changed.

Fig 2:

What is star in blot? Put in legend.

We apologize for this mistake. A star, indicating unspecific bands, is specified now in the figure legend.

A and B: denote on the Figure which antibody was used for IP and which for IB and spell this out in the legend as well. As it is presented now, it is confusing to follow.

Done.

Fig 3:

Why were the chimera complemented cells not used for IP experiments to IP Rop5?

The anti ROP5 antibody 3E2 used in this study is isoform specific. We have conclusively demonstrated in Suppl. Fig. 6 that it detects ROP5A and ROP5B but not ROP5C by Western blot analysis. Therefore, co-immunoprecipitation experiments in T17 cells complemented with *Irgb2-b1*_{Chimera} are not informative because *Irgb2-b1*_{Chimera} is associated only with

ROP5C.

E: Rather than just presenting intensity of loaded PVs, %age of loaded PVs should also be counted (like in Fig 1). Essentially, the authors just changed their read-out here. It is very different to assess the number of PVs coated versus the thickness of the coat. State why these different read-outs are used.

Recruitment of Irgb2-b1 to the *T. gondii* PVM is mediated by binding to ROP5. We show that Irgb2-b1_{CIM} is associated mainly with ROP5B but to a weaker extent also with ROP5C (Fig. 2C and D). Irgb2-b1_{Chimera} on the other hand only binds ROP5C (Fig. 3C and D). When we determine frequencies of IRG protein positive vacuoles, differences in IRG protein intensities at these vacuoles are not considered. Both, Irgb2-b1_{CIM} and Irgb2-b1_{Chimera}, are associated with ROP5 and we did not expect to find significant differences for numbers of positive vacuoles. Therefore, differences in Irgb2-b1_{CIM} and Irgb2-b1_{Chimera} loading have been defined by intensity measurements. But the reviewer is certainly right, all these considerations are not presented. We now show the mean number of Irgb2-b1_{CIM} and Irgb2-b1_{Chimera} positive vacuoles that have been determined in three independent biological triplicates (Fig. 3F). As expected, no significant differences could be observed.

Fig 4:

A: How many biological repeats of this assay were performed? Is this a representative of 3?

For every PCA assay (Fig. 2D, Fig. 3C and Fig. 4A) the mean of three biologically independent experiments is shown. All statistical methods are mentioned in the respective figure legends and defined in Material & Methods.

D: Again, this data has to be presented as the pooled data from 3 biological replicates in order to generate any statistics. Like this, it is essentially raw data of two repeats presented in a very confusing way.

We now show the mean of at least three biological triplicates for every experiment and calculated the statistical significances.

Fig 5:

A and B: This experiment should be conducted three times and not just once with 5 mice.

We now show data from three independent experiments with 25 CIM and 15 BL/6 mice in total (VAND, Fig. 5A) and from two independent experiments with 19 CIM and 10 BL/6 mice in total (AS28, Fig. 5B).

C: Are these biological triplicates or technical triplicates?

These are technical triplicates of an uracil incorporation assay that have been obtained in the laboratory of Jonathan Howard in Cologne before he translocated his laboratory to Lisbon at the IGC and we started our own group at the University Medical Center in Freiburg. Unfortunately, repetition of these analyses is not possible because the technical equipment is no longer available. We now removed these data from the manuscript because it represents only a single experiment in technical triplicates and it is currently impossible to obtain additional data for statistical analysis that is required for publication in Nature Communications.

Figure legend says MOI 1 or 3 – which one is shown in the Figure?

The Figure shows the experiment with MOI 1. Like we stated above, these data are now removed from the manuscript.

Fig 6:

Why is 200U/ml IFN γ now used? Before the amount of IFN γ was lower.

The amount of IFN γ for all these analyses (in Fig. 6 we determine protein intensities at the PVM) is always 200 U/ml. *T. gondii* replication on the other hand is always done with 100 U/ml IFN γ . Only in Suppl. Fig. 4, 10 U/ml IFN γ is used in addition. This is specified in the respective figure legends.

Figure legend A: "... protein intensities are reduced..." That is compared to what? We are thankful that this reviewer draw our attention to this. We apologize, it is a clear mistake. We corrected the sentence to "p(T108)Irga6_{CIM} protein intensities are increased at VAND-derived vacuoles."

Again, do not show two representative experiments, but rather the pooled biological triplicate repeats.

We now show the mean of at least three biological triplicates for every experiment and calculate the statistical significances.

Reviewers' comments:

Reviewer #1 (Remarks to the Author):

The authors have worked extremely hard to address my concerns and have dramatically altered the manuscript from the original submission. their thoroughness is much appreciated. This is an important paper and one that I think makes great headway on a long standing hypothesis that ROP5 paralogs have co-evolved with divergent host targets in distinct parasite lineages.

I just have 3 outstanding points:

1. Regarding the controls for the blots in Figures such as figure 2, this reviewer still feels strongly that the use of a tertiary antibody on the lysates to demonstrate that IPs do not contain unexpected protein is important to ensure that the IPs are appropriately specific. I agree that the fact that the pulldown from the ROP5 KO parasites does not contain as much IRG protein as that with the ROP5 locus intact, there is still some *Irgb2-b1* present in the KO pulldown. The lysate blots are important but an actin blot on these lysates would conclusively show consistency in terms of background levels of contaminating proteins in the actual pulldown samples, which is not possible from the data shown. I think this is particularly important for pulldowns of proteins that localize to the PVM, as others have done (such as figure 2A or 3B in Kelly et al., *MSphere* 2/3/e00183-17), where they consistently used anti GAPDH antibodies to demonstrate specificity of the IPs.
2. Regarding the inclusion of ROP18 in the model for CIM mice (fig 7) in addition to the changes in the figure legend provided the authors should still alter the figure itself to indicate the uncertainty of the ROP18 interaction (and the VAND/GRA7 interaction that they mentioned as well). This locus is very complicated and the *rop* family is extremely complicated and it is not out of the realm of possibility that diff. *rop5* paralogs have different affinities for *rop18/gra7* or any other binding partners.
3. Regarding the old world/new world discussion: (now line 469): may I just suggest a different word than "implausible" since this is speculative? Something more in line with the "more likely" statement following it? Such as it is unlikely or less likely or "would be surprising"?

Reviewer #2 (Remarks to the Author):

In the present process, the authors minimally but sufficiently responded to this reviewer's comments. Especially, quality and statistics of the data are dramatically improved.

Reviewer #3 (Remarks to the Author):

The authors have addressed all my comments satisfactorily. The manuscript is much improved and I fully support its publication in its current form.

Responses to referees' comments

Our responses are interdigitated with the referee comments, in a different font and color.

Reviewer #1 (Remarks to the Author):

The authors have worked extremely hard to address my concerns and have dramatically altered the manuscript from the original submission. Their thoroughness is much appreciated. This is an important paper and one that I think makes great headway on a long standing hypothesis that ROP5 paralogs have co-evolved with divergent host targets in distinct parasite lineages.

I just have 3 outstanding points:

1. Regarding the controls for the blots in Figures such as figure 2, this reviewer still feels strongly that the use of a tertiary antibody on the lysates to demonstrate that IPs do not contain unexpected protein is important to ensure that the IPs are appropriately specific. I agree that the fact that the pulldown from the ROP5 KO parasites does not contain as much IRG protein as that with the ROP5 locus intact, there is still some Irgb2-b1 present in the KO pulldown. The lysate blots are important but an actin blot on these lysates would conclusively show consistency in terms of background levels of contaminating proteins in the actual pulldown samples, which is not possible from the data shown. I think this is particularly important for pulldowns of proteins that localize to the PVM, as others have done (such as figure 2A or 3B in Kelly et al., *MSphere* 2/3/e00183-17), where they consistently used anti GAPDH antibodies to demonstrate specificity of the IPs.

We cannot follow Reviewer #1's argument about this matter. The data shown in Fig. 2 offer three different views of the lysates that all show the RH Δ rop5-infected lysate is quantitatively essentially identical to the RH Δ hxgprt-infected lysate:

1. In the pellets Fig. 2A (left hand panel), and in the lysate blot developed with anti-ROP (middle panel), the presumably non-specific band has the same intensity in all 4 tracks. In this context it plays the same role as an antibody against actin or tubulin or GAPDH as suggested by the reviewer.
2. In the lysates (middle panel and right hand panel), the intensity of GRA7 is nearly the same between RH Δ rop5 and RH Δ hxgprt showing the levels of infection of the two cells populations was nearly identical.
3. The level of Irgb2-b1 in the RH Δ hxgprt lysate is identical to the level of Irgb2-b1 in the RH Δ rop5 lysates (right hand panel)

We do not see any argument against the data that would require the use of another "irrelevant" antibody as proposed by the reviewer. Such an experiment will not yield a meaningful outcome that helps to improve the manuscript. The levels of a background band as well as two of the relevant molecules in the experiment, GRA7 and Irgb2-b1 are clearly essentially identical in the two lysates.

However, if the Reviewer insists, we would be happy to repeat the respective experiment with additional controls.

2. Regarding the inclusion of ROP18 in the model for CIM mice (fig 7) in addition to the changes in the figure legend provided the authors should still alter the figure itself to indicate the uncertainty of the ROP18 interaction (and the VAND/GRA7 interaction that they mentioned as well). This locus is very complicated and the rop family is extremely complicated and it is not out of the realm of possibility that diff. rop5 paralogs have different affinities for rop18/gra7 or any other binding partners.

We altered the figure legend and the figure itself (Figure 7). We think that these changes make unambiguously clear that the respective interactions within the VAND kinase complex await further experimental validation.

3. Regarding the old world/new world discussion: (now line 469): may I just suggest a different word than “implausible” since this is speculative? Something more in line with the “more likely” statement following it? Such as it is unlikely or less likely or “would be surprising”?

We did like the Reviewer suggested and changed the word „implausible“ to „unlikely“ (line 469).

Reviewer #2 (Remarks to the Author):

In the present process, the authors minimally but sufficiently responded to this reviewer's comments. Especially, quality and statistics of the data are dramatically improved.

Reviewer #3 (Remarks to the Author):

The authors have addressed all my comments satisfactorily. The manuscript is much improved and I fully support its publication in its current form.